# Dynamic decision policy reconfiguration under outcome uncertainty

**Krista Bond**[1,2,3]*, **Kyle Dunovan**[1], **Alexis Porter**[4], **Jonathan E Rubin**[2,5], **Timothy Verstynen**[1,2,3,6]*

[1]Department of Psychology, Carnegie Mellon University, Pittsburgh, United States; [2]Center for the Neural Basis of Cognition, Pittsburgh, United States; [3]Carnegie Mellon Neuroscience Institute, Pittsburgh, United States; [4]Department of Psychology, Northwestern University, Evanston, United States; [5]Department of Mathematics, University of Pittsburgh, Pittsburgh, United States; [6]Department of Biomedical Engineering, Carnegie Mellon University, Pittsburgh, United States

**Abstract** In uncertain or unstable environments, sometimes the best decision is to change your mind. To shed light on this flexibility, we evaluated how the underlying decision policy adapts when the most rewarding action changes. Human participants performed a dynamic two-armed bandit task that manipulated the certainty in relative reward (conflict) and the reliability of action-outcomes (volatility). Continuous estimates of conflict and volatility contributed to shifts in exploratory states by changing both the rate of evidence accumulation (drift rate) and the amount of evidence needed to make a decision (boundary height), respectively. At the trialwise level, following a switch in the optimal choice, the drift rate plummets and the boundary height weakly spikes, leading to a slow exploratory state. We find that the drift rate drives most of this response, with an unreliable contribution of boundary height across experiments. Surprisingly, we find no evidence that pupillary responses associated with decision policy changes. We conclude that humans show a stereotypical shift in their decision policies in response to environmental changes.

**\*For correspondence:**
kbond@andrew.cmu.edu (KB);
timothyv@andrew.cmu.edu (TV)

**Competing interest:** The authors declare that no competing interests exist.

## Editor's evaluation

The authors conducted an impressive study investigating dynamic adjustments in decision policies as a function of two types of uncertainty: decision conflict and volatility (change point probability). They combine learning model parameters with drift diffusion modeling to assess how the policy (as a combination of drift rate and threshold) varies with uncertainty and also test how these adjustments relate to the LC-NE system via pupil diameter. This work is impressive and will certainly be of interest to many.

## Introduction

'Should I stay or should I go?' refers not only to an iconic 1980s punk anthem but also the fundamental dilemma all animals face in uncertain or unstable environments. Should someone buy coffee from the cafe that serves their favorite roast or try the new cafe that opened down the street? If their favorite drink is bitter one day, is that a sign to switch to a new blend or is one subpar experience inadequate to prompt a switch? Ultimately, these decisions converge to a single predicament: whether we choose an action that we believe is likely to produce desirable results (i.e. exploit) or risk choosing another action that is less certain, on the chance that it will produce a more positive outcome (i.e. explore) (*O'Reilly, 2013*). Ultimately, this is the problem of knowing when to change your mind.

**Figure 1.** Dynamic decision policy reconfiguration. (**A**) The degree of conflict and volatility shifts the optimal balance between exploration and exploitation. (**B**) The drift diffusion model. (**C**) Accuracy (probability that left choice selected is selected; P(L)) as a function of coordinated changes in the rate of evidence accumulation (v) and the amount of information needed to make a decision, or the boundary height (a). (**D**) Reaction time as a function of changes in the rate of evidence accumulation and the boundary height. (**E**) Decision policy reconfiguration.

The shift of a decision policy from exploratory to exploitative states is driven by environmental context. To illustrate this, *Figure 1A* shows what happens when a simple reinforcement learning (RL) agent tries to maximize reward in a dynamic variant of the two-armed bandit task (*Sutton and Barto, 1998*; see Materials and methods). Here, the relative difference in reward probability for the two actions (conflict) and the frequency of a change in the optimal action (volatility) were independently manipulated. For each level of conflict and volatility, a set of tabular Q-learning (*Sutton and Barto, 1998*) agents played the task with learning rate held constant while the degree of randomness of the selection policy ($\beta$ in a Softmax function) varied. The agent that returned the most rewards was identified as the agent with the best exploration-exploitation balance. Increasing either form of uncertainty led to selecting agents with more random or exploratory selection policies (*Figure 1A*). As the value of the optimal choice decreases relative to the value of a suboptimal choice (conflict increases), the learner exploits what she already knows. Action values grow unstable (volatility increases) when the clarity of the optimal choice is constant (constant conflict), and the learner is biased toward exploration (*Bland and Schaefer, 2012*). As these two forms of uncertainty change together, the gradient of action selection strategy also changes.

Knowing *how* decision policies shift in the face of dynamic environments requires looking at the algorithmic properties of the policy itself. One popular set of algorithms for describing the dynamics of decision making are accumulation-to-bound processes like the drift-diffusion model (DDM; *Ratcliff, 1978*). The normative form of the DDM proposes that a decision between two choices is described by a noisy accumulation process that drifts toward one of two decision boundaries at a specific rate (*Figure 1B*). Two parameters of this model are critical in determining the degree of randomness of a selection policy: the rate of evidence accumulation (drift rate; $v$) and the amount of information required to make a decision (boundary height; $a$). For example, decreasing the drift rate and increasing the boundary height leads to more random decisions (*Figure 1C*), with the speed of these

decisions depending on the ratio of the two parameters (*Figure 1D*). Thus, exploratory policies can result in either fast or slow decisions, depending on the relative configuration of drift rate and boundary height.

Are the parameters that govern accumulation of evidence for decision making modifiable? Previous modeling work has shown that the parameters of a DDM process can be modulated by feedback signals and choice history (*Pedersen et al., 2017*; *Ratcliff and Frank, 2012*; *Dunovan and Verstynen, 2019*; *Dunovan et al., 2019*; *Mendonça et al., 2020*; *Urai et al., 2018*) with different mechanisms for adapting the drift rate and the boundary height. In value-based decision-making tasks where the statistics of sensory signals are equivalent for all actions, drift rate fluctuations appear to track the relative value of an action or the value difference between actions (*Dunovan et al., 2019*; *Mikhael and Bogacz, 2016*; *Bariselli et al., 2019*; *Rubin et al., 2021*). In contrast to value estimation, selection errors in this context have been linked to changes in the boundary height (*Forstmann et al., 2008*; *Forstmann et al., 2010*; *Bogacz et al., 2010*; *Herz et al., 2016*; *Herz et al., 2017*; *Dunovan et al., 2019*; *Dunovan and Verstynen, 2019*) and internal estimates of environmental change (*Nassar et al., 2010*; *Wilson and Niv, 2011*; *Nassar et al., 2012*; *Behrens et al., 2007*).

Given the adaptive sensitivity of the drift rate and the boundary height to value estimation and selection, respectively, these decision parameters define unique states on a surface of fast or slow and exploratory or exploitative decision policies. These policies, in turn, adaptively reconfigure based on current environmental feedback signals by modulating value estimation and the rate of selection errors (*Figure 1E*). Agents can move along the surface of decision policies, from exploitative states (bright colors, *Figure 1E*) to different types of exploratory states (darker colors, *Figure 1E*), as they commit a greater number of selection errors prompted by change in action-outcome contingencies. As the system relearns properties of the environment, the decision policy migrates along the surface to return to an exploitative state until a change occurs again.

One plausible neural mechanism for this migration along the surface of selection policies is the locus coereleus norepinephrine (LC-NE) system, which has been linked to adaptive behavioral variability in response to uncertainty (*Urai et al., 2017*; *Dayan and Yu, 2006*; *Bouret and Sara, 2005*). The LC-NE system has two distinct modes (*Aston-Jones and Bloom, 1981*) that map onto distinct decision states (*Aston-Jones and Cohen, 2005*). In the phasic mode, a burst of LC activity results in a global, temporally precise release of NE. This increases the gain on cortical processing and encourages exploitation. In the tonic mode, NE is released without the temporal precision of the phasic mode, increasing baseline NE (*Aston-Jones and Bloom, 1981*). This encourages disengagement from the current task and facilitates exploration. The dynamic fluctuation of these two modes is thought to optimize the trade-off between the exploitation of stable sources of reward and the exploration of potentially better options (*Aston-Jones and Cohen, 2005*). Thus the LC-NE system, which can be indirectly measured by fluctuations in pupil diameter (*Aston-Jones and Cohen, 2005*; *Jepma and Nieuwenhuis, 2011*), may be a central mechanism for modulating selection policies.

We investigated the malleability of decision policies as the environment necessitates a change of mind as to what constitutes the 'best' decision. To control environmental uncertainty, we manipulated the volatility of changes in action-outcome contingencies (i.e. which of two targets returns the most rewards), as well as ambiguity in optimal choice (*conflict*), while human participants performed a dynamic variant of the two-armed bandit task (*Sutton and Barto, 2018*). We predicted that, in response to suspected changes in action-outcome contingencies, humans would exhibit a stereotyped adjustment in the drift rate and boundary height that pushes decisions from certain, exploitative states to uncertain, exploratory states and back again (*Figure 1E*). In addition, using pupillary data, we explored whether the LC-NE system covaries with shifts of the boundary height in response to a change in action outcomes to facilitate exploration, consistent with prior studies (*Keung et al., 2019*; *Murphy et al., 2014*; *Cavanagh et al., 2014*).

## Results

Across two experiments, we used a dynamic two-armed bandit task with equivalent sensory reliability across arms to independently manipulate the reward conflict and the volatility of action outcomes in order to measure how underlying decision processes respond to changes in action-outcome contingencies (see Stimuli and Procedure). Both of these experiments shared a common feedback structure. Participants were asked to select either the left or right target presented on the screen using the

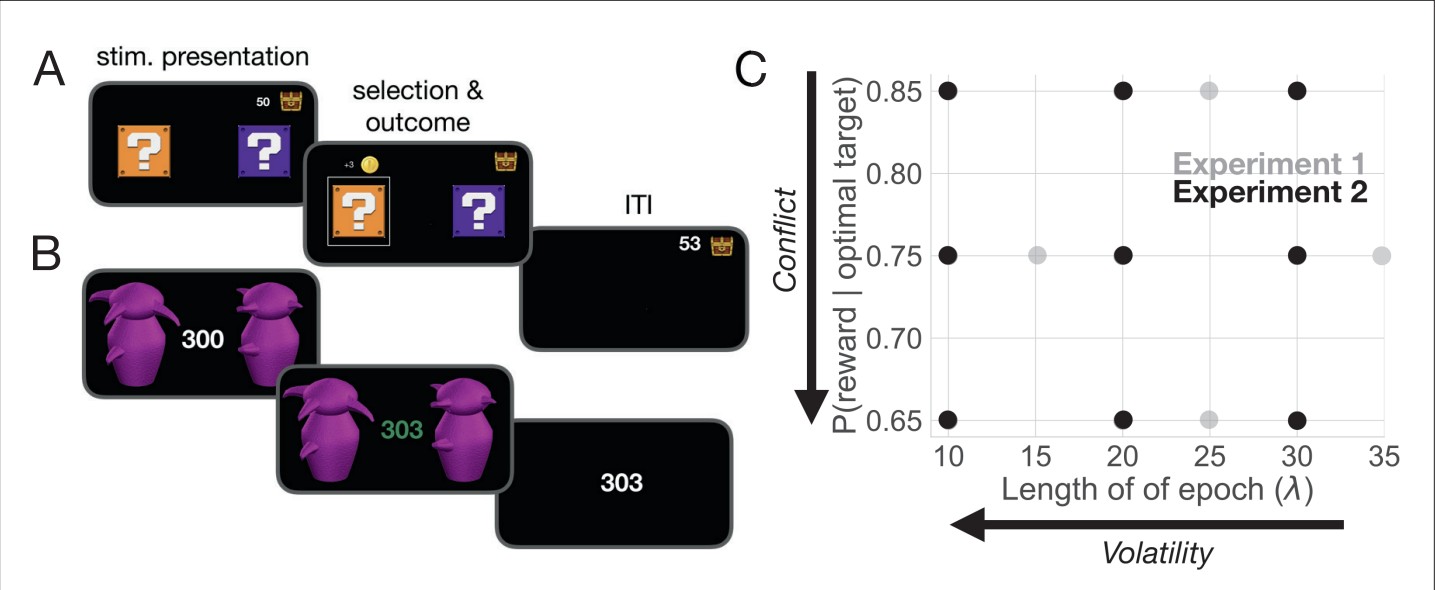

**Figure 2.** Task and uncertainty manipulation. (**A**) In Experiment 1, participants were asked to choose between one of two 'mystery boxes'. The point value associated with a selection was displayed above the chosen mystery box. The sum of points earned across trials was shown to the left of a treasure box on the upper right portion of the screen. (**B**) In Experiment 2, participants were asked to choose between one of two Greebles (one male, one female). The total number of points earned was displayed at the center of the screen. The stimulus display was rendered isoluminant throughout the task. (**C**) The manipulation of conflict and volatility for Experiments 1 (gray) and 2 (black). Each point represents the combination of degrees of conflict and volatility. Under high conflict, the probability of reward for the optimal and suboptimal target is relatively close. Under high volatility, a switch in the identity of the optimal target selection is relatively frequent.

corresponding key on a response box. Rewards were probabilistically determined for each target and, if a reward was delivered, it was sampled from a Gaussian distribution. The optimally rewarding target delivered reward with a predetermined probability ($P(optimal)$) and the suboptimal target gave reward with the inverse probability ($1 - P(optimal)$). After a delay determined by the rate parameter of a Poisson distribution ($\lambda$), the reward probabilities for the optimal and suboptimal targets would switch.

In Experiment 1, 24 participants completed four sessions (high and low conflict; high and low volatility) each composed of 600 trials. During each session, they were asked to select one of two coin boxes (Exp. 1: **Figure 2A**). The levels of conflict and volatility for all four conditions in Experiment 1 are shown as gray dots in **Figure 2C**. Experiment 2 was a replication of Experiment 1 with more extensive within-subject sampling of conflict and volatility, as well as the inclusion of pupillometry as a proxy for measuring LC-NE dynamics. In Experiment 2, participants were asked to choose between one of two Greebles (one male, one female). Each Greeble probabilistically delivered a monetary reward (Exp. 2: **Figure 2B**). Participants were trained to discriminate between male and female Greebles prior to testing to prevent errors in perceptual discrimination from interfering with selection on the basis of value estimation. Four participants completed nine sessions composed of 400 trials each, generating 3600 trials in total per subject. The levels of conflict and volatility for all nine conditions in Experiment 2 are shown as black dots in **Figure 2C**. Importantly, Experiment 2 manipulated the same forms of uncertainty as Experiment 1, but had different perceptual features and more expansively sampled the space of conflict and volatility. Given the similarity in design, the behavioral results for both of these experiments are presented together below.

## The influence of ambiguity and instability on speed and accuracy

We first looked at overall speed and accuracy effects in both Experiments 1 and 2. In Experiment 1, accuracy (i.e. optimal choice selection) suffered as the optimal choice grew more ambiguous, with accuracy in the low conflict condition being 1.2 times higher than what is observed in the high conflict condition (**Figure 3A**; $\beta = 1.213$, 95% CI: 1.192, 1.235, z = 21.36, p < 2e-16). In contrast, increasing conflict had no observable impact on overall reaction times (**Figure 3A**; $\beta = -6.902e^{-}5$, 95% CI: –0.002, 0.002, t = –0.06, p = 0.951). As expected, participants also became less accurate as

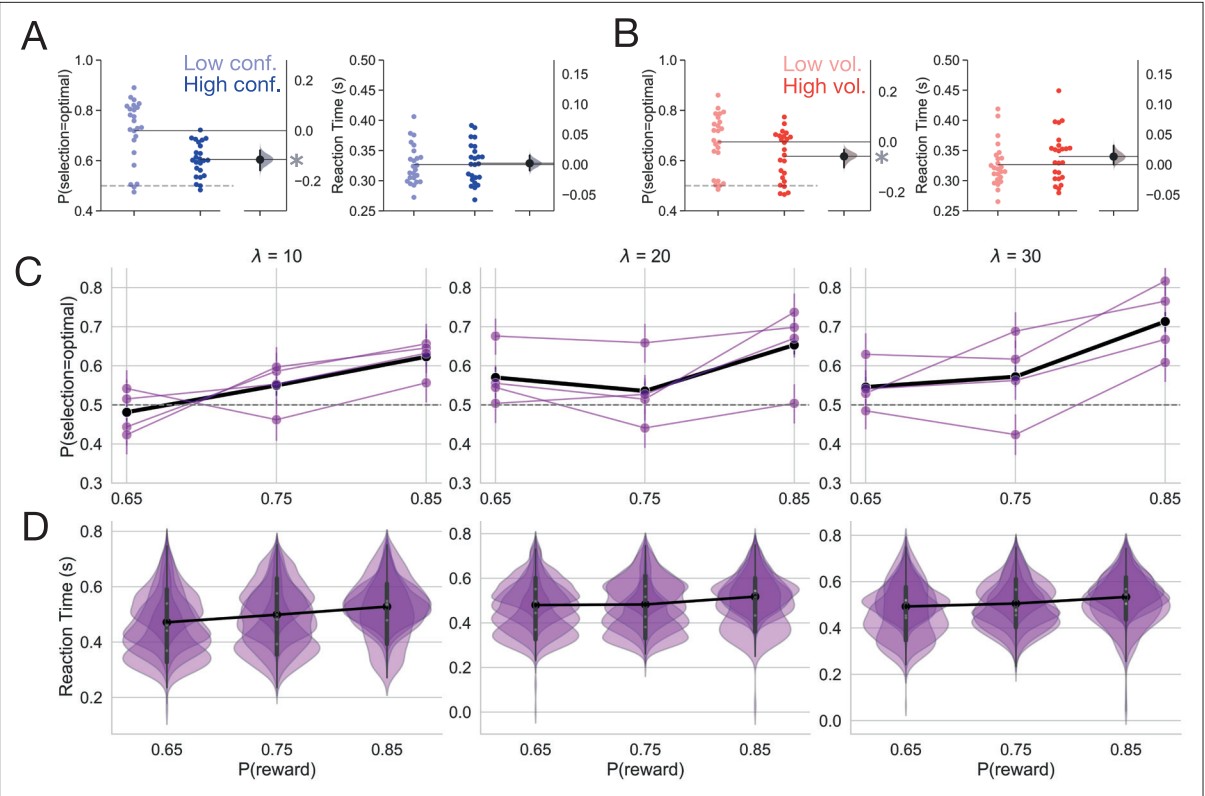

**Figure 3.** Behavior. (**A**) Mean accuracy and reaction time for the manipulation of conflict in Experiment 1. (**B**) Mean accuracy and reaction time for the manipulation of volatility in Experiment 1. Each point represents the average for a single subject. The distribution to the right represents the bootstrapped uncertainty in the mean difference between conditions (high conflict or high volatility subtracted from low conflict or low volatility). Distributions with 95% CIs that do not encompass 0 are marked with an asterisk. (**C**) Mean accuracy for Experiment 2. Each purple line represents a subject. The black line represents the mean accuracy calculated across subjects. (**D**) Reaction time distributions for each subject for Experiment 2. The black line represents the mean reaction time calculated over subjects. Error bars indicate a bootstrapped 95% confidence interval. For panels C and D, $\lambda$ values shown above each plot specify the average period of optimal choice stability and the probability of reward shown on the x-axis specifies the degree of conflict. Means are calculated over all trials.

the instability of action outcomes (i.e. volatility) grew (**Figure 3B**; $\beta = 0.092$, 95% CI: 0.077, 0.111, z = 10.36, p < 2e-16). Under volatile conditions, participants also took slightly longer to make a decision ($\beta = -0.012$, 95% CI: −0.015,−.010, t = −10.80, p < 2e-16); however, while this effect on reaction times was statistically reliable, the impact of volatility on reaction times was weak (increasing volatility increased reaction time by ~13 ms on average; **Figure 3B**).

Experiment 2 served as a high powered test of whether the effects we observed in Experiment 1 were replicable at the within-subject level. Because Experiment 2 independently manipulated conflict and volatility, we were able to test whether conflict and volatility interacted to affect behavior. We found similar effects of conflict and volatility on accuracy as we observed in Experiment 1 (**Figure 3C**). Accuracy increased as conflict decreased (i.e. as the probability of reward increased; $\beta$=0.223, 95% CI = 0.189,0.256, z = 12.757, p<2e-16). As the environment grew less volatile, accuracy increased ($\beta = 0.101$, 95% CI = 0.066,0.14, z = 5.828, p = 5.6e-09). We did not observe an interaction of conflict and volatility on accuracy ($\beta = 0.024$, 95% CI = −0.013, 0.058, z = 1.364, p = 0.173).

However, we did find that conflict and volatility interacted to affect reaction time (RT; $\beta$=−0.002, 95% CI = −0.004, −0.001, t = −3.084, p = 0.002), with a linear increase in reaction time as the environment grew less volatile and conflict was highest (when $p(r) = 0.65$ as a function of $\lambda$; see **Figure 3D** for RT distributions). When conflict was moderate ($p(r) = 0.75$) or low ($p(r) = 0.85$), volatility had a nonlinear effect on RTs. Here, reaction times decreased when volatility was moderate ($\bar{RT} = 0.483$ when $\lambda$=20 and $p(r) = 0.75$ when $\lambda$=20 and $p(r) = 0.85$). Reaction times increased to approximately the same extent within moderate or low conflict conditions when volatility was high ($\bar{RT} = 0.499$ when $\lambda$=10 and $p(r) = 75$ when $\lambda$=10 and $p(r) = 0.85$) and when volatility was low ($\bar{RT} = 0.506$ when $\lambda = 30$

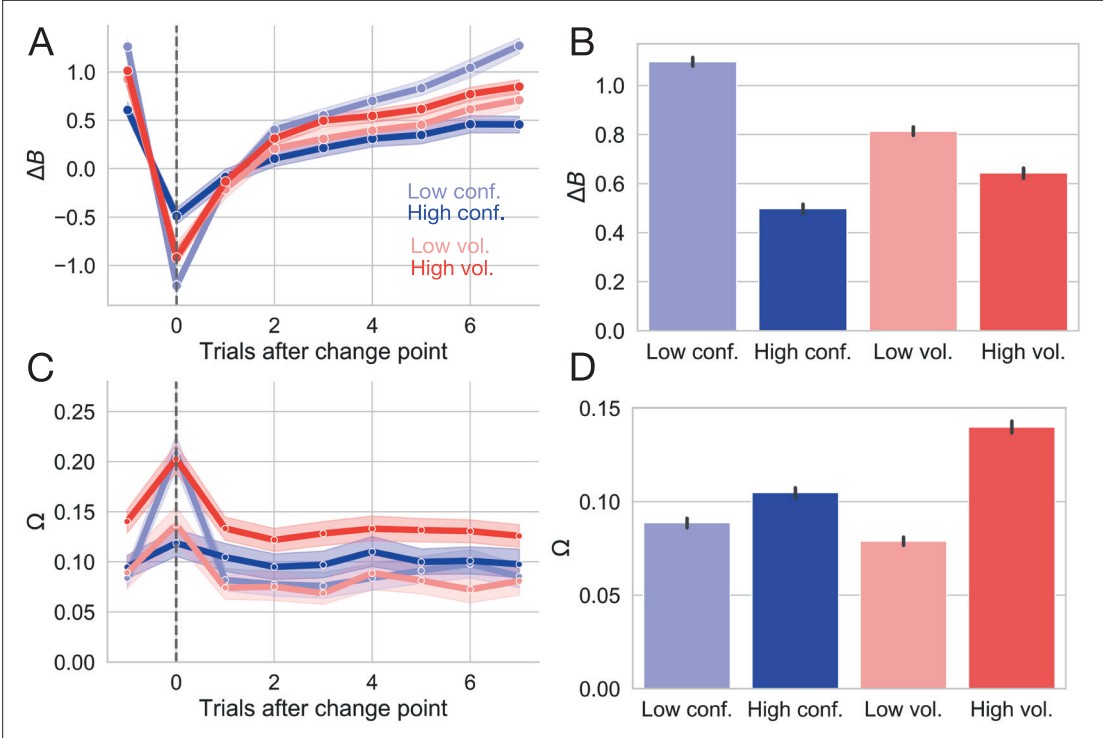

**Figure 4.** Changes in ideal observer estimates as a function of condition for Experiment 1. (**A**) Changes in the belief in the value of the optimal target ($\Delta B$) as a function of conflict and volatility over time. (**B**) Belief in the value of the optimal choice by condition and averaged over all trials. (**C**) Changes in change point probability ($\Omega$) as a function of conflict and volatility over time. (**D**) Change point probability by condition and averaged over all trials. Error bars represent 95% CIs.

and $p(r) = 0.75$ when $\lambda = 30$ and $p(r) = 0.85$), with an increase in baseline reaction times when conflict was low relative to moderate ($\bar{RT} = 0.527$ when $p(r) = 0.85$ when $p(r) = 0.75$; see **Appendix 1—figure 1** for interaction visualization).

At the gross level, over all trials within an experimental condition, increasing the ambiguity of the optimal choice (conflict) and increasing the instability of action outcomes (volatility) decreases the probability of selecting the optimal choice. Reaction time effects were inconsistent, with a negligible effect of volatility in Experiment 1. Experiment two revealed that volatility and conflict interact to influence reaction times in complex ways. However, because trials where action-outcome contingencies change are so infrequent, even under high volatility conditions, these overall effects on speed and accuracy may be masking more subtle behavioral dynamics in response to feedback changes. We adopt a more focal, model-based analysis in the next section to clarify these peri-change point dynamics.

## Tracking estimates of action value and environmental volatility

We calculated trial-by-trial estimates of two ideal observer parameters of environmental states (see Cognitive model for calculation details; **Nassar et al., 2010**; **Vaghi et al., 2017**). Belief in the value difference ($\Delta B$) reflects the difference between the learned values of the optimal and suboptimal targets. For ease of interpretation, we refer to the converse of belief as doubt, such that when belief decreases doubt increases. $\Delta B$ thus reflects a local estimate of uncertainty regarding the choices themselves. To capture the estimated probability of fundamental shifts in action values, we calculated how often the same action gave a different reward (change point probability; $\Omega$). Here, $\Omega$ reflects a global estimate of uncertainty in the environment, specifically the uncertainty in response contingencies. We used the data from Experiment 1 to assess how well these learning estimates captured our imposed manipulations, and observed similar results in Experiment 2 (**Appendix 1—figure 4**).

In Experiment 1, we observed a sharp decrease in $\Delta B$ after a switch in action outcomes and a gradual return to asymptotic values (**Figure 4A**) with a decreased difference in reward probability

resulting in increased doubt (*Figure 4B*; $\beta = 0.216$, 95% CI:0.206, 0.224, t = 46.24, p < 2e-16). As expected, less volatile conditions allowed the learner to more fully update her belief in the value of the optimal choice over all trials ($\beta = 0.058$, 95% CI:0.050, 0.068, t = 12.32, p < 2e-16), though to a smaller degree than low conflict conditions allowed (see *Figure 4B*). Increasing volatility resulted in a sharp increase in the estimate of $\Omega$ at the onset of a change point with a quick return to a baseline estimate of change (*Figure 4C*). Notably, this estimate of $\Omega$ was more sensitive to change points when conditions were relatively volatile, with a more pronounced peak in response to a change under high volatility conditions than under low volatility conditions (*Figure 4C*). Correspondingly, over all trials, $\Omega$ was higher under more volatile conditions (*Figure 4D*, $\beta = -0.022$, 95% CI:−0.023,−0.020, t = −30.74, p < 2e-16) indicating sensitivity to the increased frequency of action outcome switches in the reward schedule.

When the identity of the optimal choice was clear (i.e. when conflict was low), the estimate of $\Omega$ was more sensitive to the presence of a true change point than when the optimal choice was ambiguous (i.e. when conflict was high) (*Figure 4C and D*). This observation is consistent with the idea that increasing the difficulty of value estimation and, thereby, the assignment of value to a given choice also impairs change point sensitivity. Interestingly, increasing conflict nevertheless resulted in a net increase in $\Omega$ calculated over all trials (*Figure 4D*; $\beta$=−0.006, 95 %CI:−0.007,−0.004, t = −8.64, p < 2e-16), likely because higher conflict conditions increased the baseline estimate of change instead of enhancing sensitivity to true change points (see change point response and relative baseline values for the high conflict condition in *Figure 4C*). Here, the system conservatively over-estimates the volatility of action outcomes, assuming a slightly greater frequency of changes in the probability of reward for the optimal choice than we imposed (actual proportion of change points for high conflict condition: $0.041 \pm 0.004$; estimated $\Omega$).

Reassuringly, net change point probability was much greater when change points were more frequent (see increased $\Omega$ estimates for high volatility conditions over high conflict conditions in *Figure 4D*). These results suggest that our formulation of these ideal observer estimates adequately captures our manipulation of volatility and conflict at a continuous level.

Thus, these ideal observer parameters show a reliable response to a change in action-outcome contingencies. The difference in value belief decreases, or doubt increases, when a change point occurs and slowly recovers over the course of six to eight trials as participants learn new action-outcome contingencies. The initial drop in belief difference is deeper and the recovery time after a change point is slower in conditions with greater overall uncertainty (i.e. under high conflict and high volatility). In contrast, internal estimates that a change has occurred briefly spike at a change point, indicating that participants can reliably detect that something has changed, and quickly settle after a few trials. Interestingly, net change point probability estimates are higher in the conditions with higher uncertainty (high conflict, high volatility), likely reflecting increased vigilance for changes in those conditions. In the next section, we explore how the underlying parameters of the decision process itself respond to local changes in action-outcome contingencies.

## Different forms of uncertainty impact distinct decision processes

Our next goal was to test which decision parameters were sensitive to a change point. To this end, we estimated the change point evoked response of the boundary height $a$, drift rate $v$, non-decision time $t$, starting bias $z$, and drift criterion $dc$ for each trial surrounding the change point. To detect changes in the change-point-evoked distributions for each decision parameter, we evaluated whether the sequential distributions evoked by each trial were significantly different, beginning with the trial preceding the change point and ending three trials after the change point. For example, if the 95% CI of the $z$ distribution evoked on the trial prior to the change point overlapped with the 95% CI of the distribution evoked on the change point and so on for all successive trials considered, then we would conclude that $z$ failed to show change point sensitivity (see Hierarchical drift diffusion modeling for details). To select the model that best accounted for the data, we compared the deviance information criterion (DIC) scores (*Spiegelhalter et al., 2002*) for these models. DIC scores provide a measure of model fit adjusted for model complexity and quantify information loss. A lower DIC score indicates a model that loses less information. Here, a difference of ≤ two points from the lowest-scoring model cannot rule out the higher scoring model; a difference of 3–7 points suggests that the higher scoring model has considerably less support; and a difference of 10 points

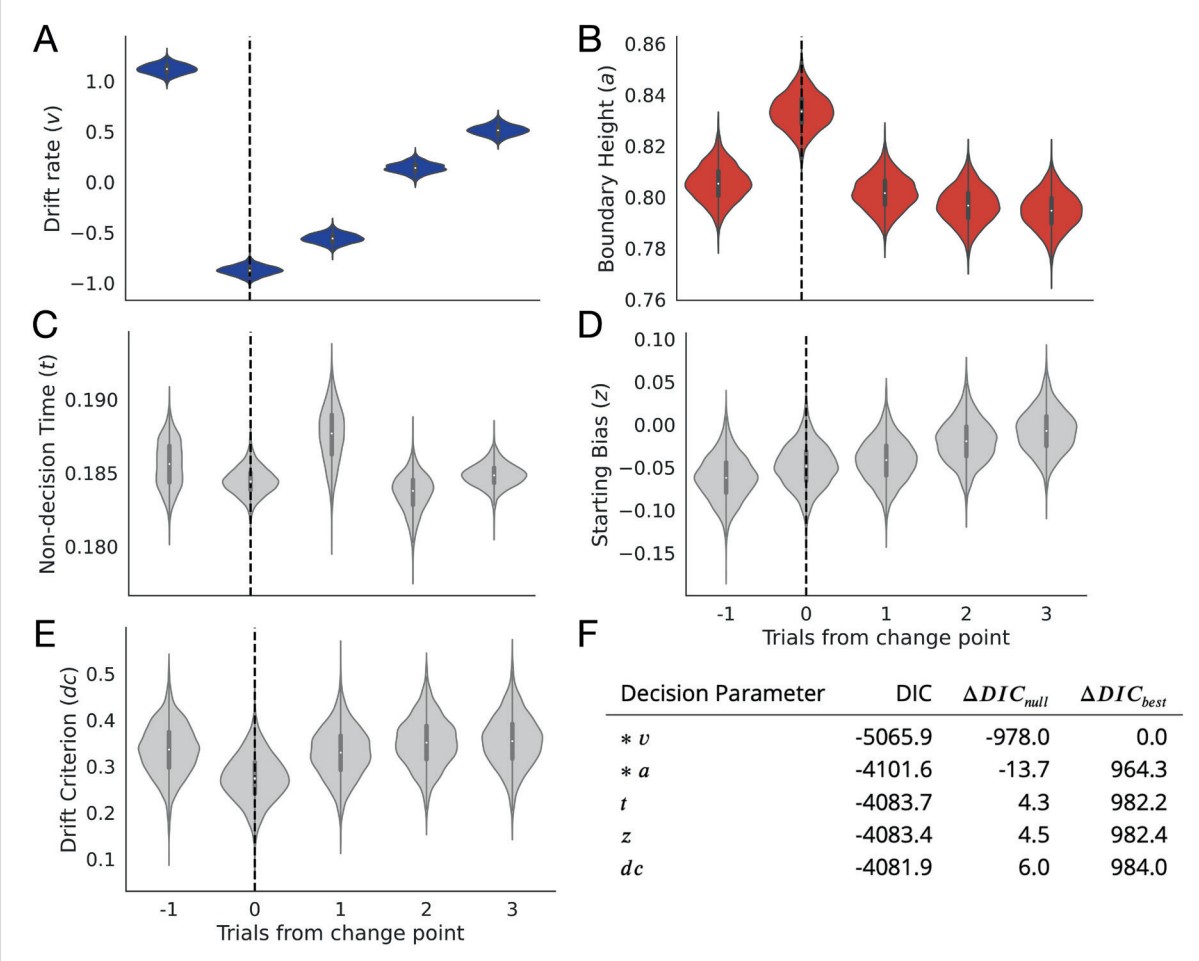

**Figure 5.** Change point sensitivity of underlying decision processes. Posterior distributions for each decision parameter are shown for the trial prior to a change point to three trials after the change point. (**A**) The drift rate. (**B**) The boundary height. (**C**) Non-decision (onset) time. (**D**) Starting bias. (**E**) Drift criterion. (**F**) Degree of fit to observational data as information loss. The models that lost the least information are marked with an asterisk.

suggests essentially no support for the higher scoring model (*Spiegelhalter et al., 2002*; *Burnham and Anderson, 1998*).

Under this analysis, we found that only the boundary height and drift rate showed change point sensitivity as defined above. The drift rate showed a clear, persistent separation between trial-specific distributions, with a rapid decrease at the onset of the change point ($t_{-1}$ 95% CI = 1.021, 1.218; $t_0$=−0.972, −0.779) and a return to baseline values thereafter ($t_1$ = −0.656,,−0.46; $t_2$=0.039, 0.241; $t_3$=0.411, 0.616; *Figure 5A*). The boundary height showed a transient response to the change point, spiking ($t_{-1}$ 95% CI = 0.792, 0.819; $t_0$=0.820, 0.847) and then dropping to baseline levels ($t_1$ = 0.789, 0.815; $t_2$=0.783, 0.811; $t_3$=0.780, 0.808; *Figure 5B*).

The remainder of the decision parameters showed no change point sensitivity. Non-decision time showed no clear response ($t_{-1}$ 95% CI = 0.183, 0.188; $t_0$=0.183, 0.186; $t_1$=0.184, 0.191; $t_2$=0.181, 0.186; $t_3$=0.183, 0.186; *Figure 5C*) along with the starting bias ($t_{-1}$ 95% CI = −0.112,,−0.01; $t_0$=−0.098, 0.002; $t_1$=−0.090, 0.008; $t_2$=−0.069, 0.032; $t_3$=−0.055, 0.045; *Figure 5D*) and the drift criterion ($t_{-1}$ 95% CI = 0.229, 0.439; $t_0$=0.175, 0.374; $t_1$=0.223, 0.435; $t_2$=0.245, 0.458; $t_3$=0.244, 0.464; *Figure 5E*).

Further, models fitting drift rate and boundary height lost the least null-model-adjusted information relative to models of the change-point-evoked response for the other parameters, showing that a change-point-evoked decrease in drift rate and spike in the boundary height best accounted for our observational data in comparison to all alternatives ($\Delta DIC_{null}$ for $v$ = −978 and $\Delta DIC_{null}$ for $a$ = −13.7; see *Figure 5F*).

**Table 1.** Model comparison for Experiments 1 and 2.

Roman numerals refer to a given model, as defined by the mapping between the ideal observer estimates and decision parameters in the first two columns. The left panel shows the deviance information criterion (DIC) scores for the set of models considered during the model selection procedure for Experiment 1. The right panel shows the DIC scores for the equivalent model selection analysis for Experiment 2, with a model estimated for each of four subjects. Values shown represent the mean and standard deviation computed over subjects. Note that the raw DIC values for each of the subjects in Experiment 2 are included in *Appendix 3—table 1*. The column labeled DIC gives the raw DIC score, $\Delta \text{DIC}_{null}$ lists the change in model fit from an intercept-only model (the null-adjusted fit), and $\Delta DIC_{best}$ provides the change in null-adjusted model fit from the best-fitting model. The best performing model is denoted by an asterisk, with equivocal best cases marked by a tilde.

**Experiment 1**

|  | $\Delta B$ | $\Omega$ | DIC | $\Delta \text{DIC}_{null}$ | $\Delta DIC_{best}$ |
|---|---|---|---|---|---|
| *I | *v* | *a* | −18643.9 | −2698.0 | 0.0 |
| II | *a* | *v* | −16265.6 | −319.7 | 2378.3 |
| III | – | *v* | −16180.5 | −234.7 | 2463.3 |
| IV | *v* | – | −18630.8 | −2684.9 | 13.1 |
| V | – | *a* | −15949.20 | −3.4 | 2694.7 |
| VI | *a* | – | −16032.8 | −87.0 | 2611.1 |
| VII | – | – | −15945.8 | 0.0 | 2698.0 |

**Experiment 2**

|  | $\Delta B$ | $\Omega$ | $\Delta \text{DIC}_{null}$ | $\Delta \text{DIC}_{best}$ |
|---|---|---|---|---|
| *~I | *v* | *a* | −90.3 ± 71.7 | 1.0 ± 0.8 |
| II | *a* | *v* | −7.6 ± 13.1 | 83.8 ± 60.5 |
| III | – | *v* | −8.5 ± 13.1 | 82.9 ± 61.4 |
| *~IV | *v* | – | −90.8 ± 71.0 | 0.5 ± 1.1 |
| V | – | *a* | 0.3 ± 2.5 | 91.6 ± 70.6 |
| VI | *a* | – | 0.95 ± 1.4 | 92.3 ± 70.9 |
| VII | – | – | 0 ± 0 | 91.3 ± 71.5 |

Given that only the drift rate and boundary height showed change point sensitivity, we next focused on how those two parameters related to internal estimates of change and conflict in both experiments. Recall that we used the ideal observer parameters $\Delta B$ and $\Omega$ as proxies for internal estimates of belief in the difference in learned target values and change point probability, respectively. This provided a continuous quantification of our manipulation of conflict and volatility (see Tracking estimates of action value and environmental volatility). Experiment 2 provided an intensively sampled within-subject test of the change-point-evoked mapping between decision processes and these ideal observer estimates.

In order to determine the nature of the mapping between the ideal observer parameters and the change-point sensitive decision parameters, we estimated single and dual-parameter models mapping $\Delta B$ and $\Omega$ and the change-point-sensitive decision parameters, drift rate and boundary height, and examined the fit of these models to our data. We found that the model mapping $\Delta B$ to drift rate and $\Omega$ to boundary height provided the best fit in Experiment 1 ($\Delta DIC_{null} = -2698.0$; left panel of *Table 1*).

To test whether this mapping was preserved in an independent data set, we performed the same model comparison procedure for Experiment 2. Because Experiment 2 followed a replication-based design, we fit a separate model to each subject to assess the replicability of the best fitting model from Experiment 1. While we found support for the model mapping $\Delta B$ to drift rate and $\Omega$ to boundary

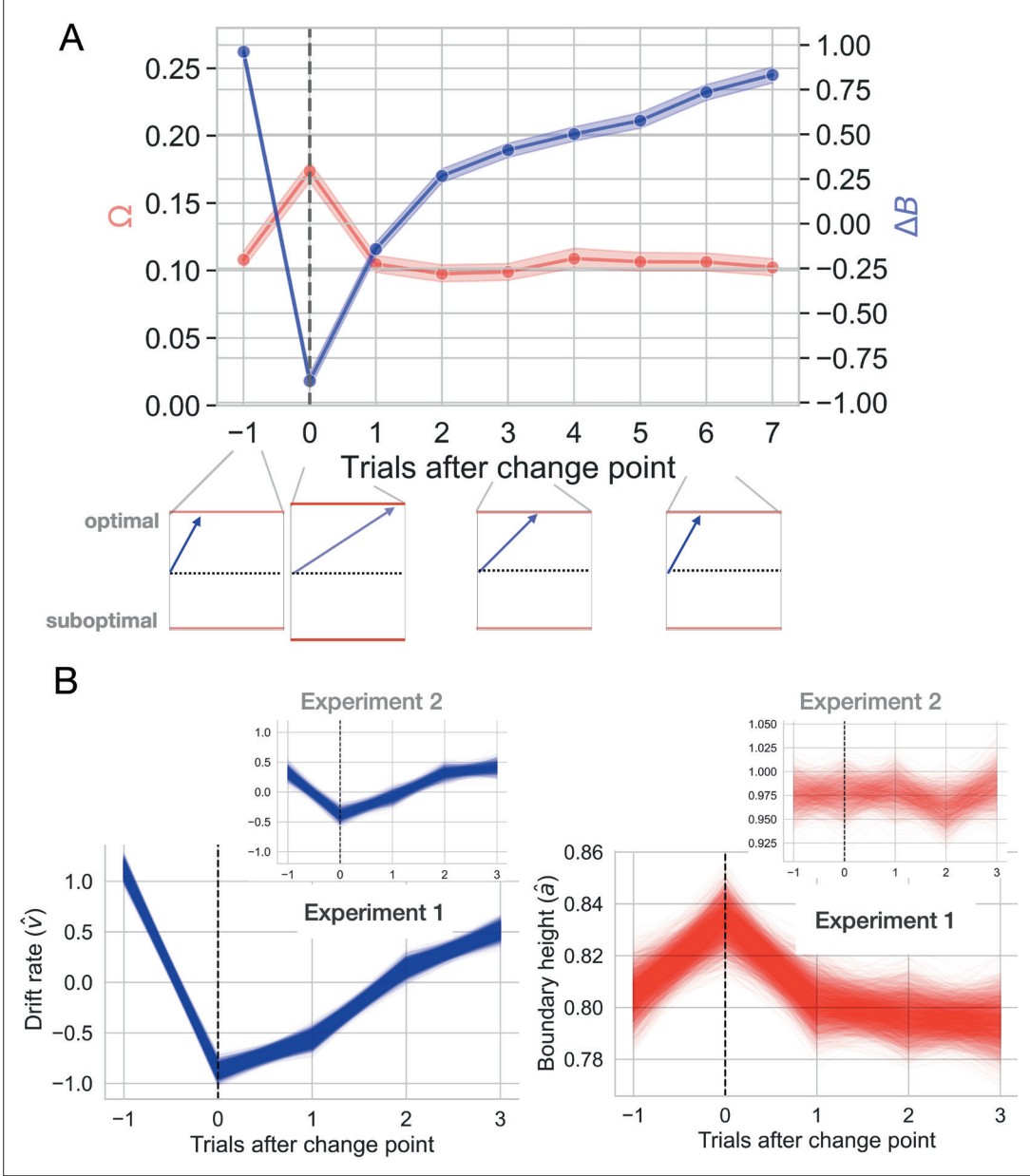

**Figure 6.** Change-point-evoked uncertainty. (**A**) Changes in ideal observer estimates of uncertainty over time and their effect on the boundary height and the drift rate. Directly after a change point, the boundary height *increases* and the drift rate slows. Over time, the boundary height returns to its baseline value and the drift rate increases. (**B**) Fitted estimates of change-point-evoked drift rate and boundary height for both experiments with 95% CIs of the posterior distributions. Inset plots represent data from Experiment 2.

height, we also found that the DIC scores for the single-parameter model mapping $\Delta B$ to $v$ alone fit the data equally well (see bottom panel of *Table 1* for summary statistics and *Appendix 3—table 1*). Altogether, this suggests that we have strong evidentiary support for a mapping between value-driven belief and drift rate (*Figure 6A*, blue). However, the support for a mapping between change point probability and boundary height (*Figure 6A*, red), while robustly present in Experiment 1, fails to appear when tested in an independent data set.

For a more granular assessment of how drift rate and boundary height respond to a change point, we quantified the change-point-evoked effect of $\Delta B$ and $\Omega$ on drift rate and boundary height, respectively, for both experiments (see Hierarchical drift diffusion modeling for details). In Experiment 1, we found that the rate of evidence accumulation, $v$, increased with the belief in the value of the

optimal choice relative to a change point ($\beta_{v\sim\Delta B} = 0.576$, 95% CI: 0.544, 0.609, empirical $p = 0.000$; *Figure 6B*, left panel). The boundary height *increased* with change point probability ($\beta_{a\sim\Omega} = 0.046$, 95% CI: 0.005, 0.088, empirical $p = 0.001$; *Figure 6B*, right panel).

Experiment two showed similar, but attenuated, results, with drift rate increasing with $\Delta B$ ($\beta_{v\sim\Delta B} = 0.112$, 95% CI: 0.016, 0.227, empirical $p = 0.004$; *Figure 6B*, inset panel on left) and an unreliable effect of $\Omega$ on boundary height ($\beta_{a\sim\Omega} = -0.036$, 95% CI: –0.155, 0.097, empirical $p = 0.282$; *Figure 6B*, inset panel on right). Therefore, as the belief in the value of the optimal choice approaches the reward value for the optimal choice, the rate of information accumulation increases. An internal estimate of change point probability weakly increases the amount of information required to make a decision, although this latter effect is less reliable.

Altogether, these results suggest a drift rate mechanism for adaptation to change that may also combine with boundary height dynamics (*Figure 6A*). However, the strength of the drift rate response weakened and the boundary height response was statistically unreliable in Experiment 2 (*Figure 6B* inset panels). When a change point is detected and the threshold for committing to a choice (*a*) responds, it shows a weak, transient increase. At the same time, the drift rate approaches zero, allowing time for the decision process to diffuse and encouraging a random selection. As the learner accrues information about the new optimal choice, the rate of information accumulation slowly recovers to asymptotic levels, with the decision process assuming a more directed path toward the choice that has accrued evidence for reward. Together, the changes in these underlying decision processes, largely driven by drift rate dynamics, point to a mechanism for gathering information in a relatively slow, unbiased manner shortly after the learner suspects she should update her valuation. We now explore these dynamics in more detail in the next section.

## Environmental instability prompts a stereotyped decision trajectory

So far, we have established that both the drift rate and the boundary height can be independently manipulated by two different estimates of environmental uncertainty with different temporal dynamics, although this effect reduces to drift rate dynamics in Experiment 2. This suggests that a change in

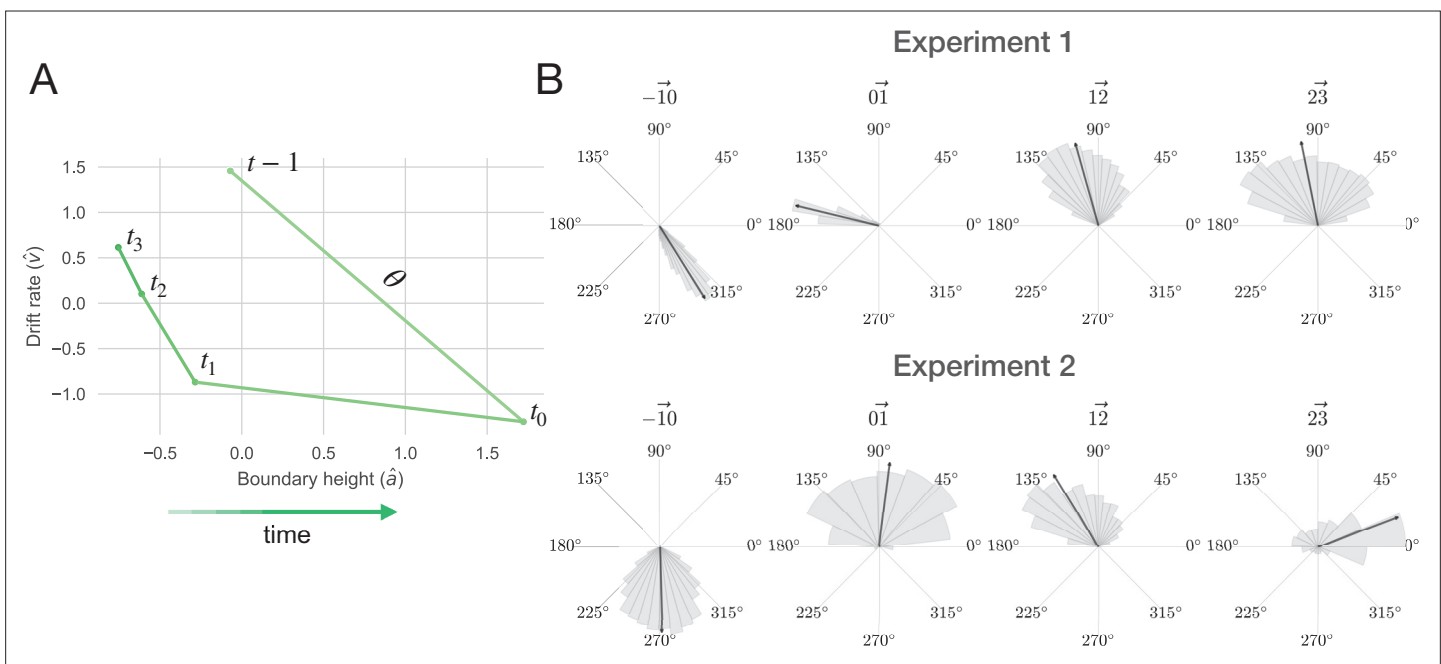

**Figure 7.** The decision surface. (**A**) Representing decision space in vector form. An angle ($\theta$) was calculated between sequential values of ($a, v$) coordinates, beginning with the trial prior to the change point. This represents subject-averaged data from Experiment 1. Note that these trajectories are z-scored. (**B**) Distributions depicting the angle between drift rate and boundary height for both Experiments 1 and 2. Each subpanel shows the distribution of angles between ($a, v$) over sequential trials, beginning with the trial prior to the change point. The area of the shaded region is proportional to the density and the arrow represents the circular mean.

action-outcome contingencies prompts a unique trajectory through the space of possible decision policies (*Figure 1E*).

To visualize this trajectory, we plot the temporal relationship between drift rate and boundary height beginning with the trial prior to the change point and ending three trials after the change point (*Figure 7A*). To clearly visualize the distribution of the change-point driven response in the relationship between drift rate and boundary height over time, we also represent the trialwise shift in these two decision variables as vectors. The trial-by-trial estimates of drift rate and boundary height were taken from the best model of the fitted change-point-evoked response and z-scored (see Different forms of uncertainty impact distinct decision processes for model selection). Then the difference between each sequential set of boundary height and drift rate coordinates, $(a, v)$, was calculated to produce a vector length. The arc tangent between these differenced values was computed to yield an angle in radians between sequential decision vectors, concisely representing the overall decision dynamics ($\theta$, *Figure 7B*; see Decision vector representation for methodological details).

For Experiment 1, following a shift in response contingencies, the navigation of this decision surface follows a stereotyped pattern. The boundary height spikes and drift rate decreases rapidly, gradually recovering and stabilizing over time (see the trial prior to the change point in *Figure 7A*). This decision trajectory is robust in Experiment 1 (*Figure 7B*, top panel).

Here, we find that the distribution of $\theta$ prior to a change point averages to ~300°, sharply changes in response to the observation of a change point (~165°) and steadily returns to values prior to the onset of a change (main panels in *Figure 7B*). One trial after the change point, drift rate sharply decreases and boundary height spikes, after which boundary height quickly recovers and drift rate steadily progresses toward its baseline value.

However, this trajectory is substantially more variable in Experiment 2, with most of the response restricted to the drift rate dimension and inconsistent trajectories along the boundary height dimension (*Figure 7B*, lower panel). Here, the distribution of $\theta$ prior to a change point averages to ~270° and shifts to ~90° with the observation of a change. In both experiments, we find that the decision trajectory quickly responds to a shift in action outcomes and also quickly recovers and stabilizes.

Having characterized the change-point-evoked trajectory through the range of decision policies, we next asked whether conditions of increased volatility and increased conflict might modify its path. To this end, we conducted a comparison of a null model with models specifying the change-point evoked response alone and this evoked response as a function of conflict and volatility. To estimate this relationship between drift rate and boundary height, we used Bayesian circular regression (*Mulder and Klugkist, 2017*). First, we tested the null hypothesis that the decision dynamics (the relationship between drift rate and boundary height; $\theta$) were solely a function of the intercept, or the average of the decision dynamics $\theta$:

$$\theta = \beta_0$$

We call this the null model.

To test the hypothesis that decision dynamics varied solely as a function of time after a switch in action-outcome contingencies, we estimated the change in $(a, v)$ coordinates ($\theta$) relative to a change point, with the time scale of consideration determined by the results of a stability analysis from Experiment 1 (see Model proposals and evaluation; *Appendix 1—figure 5*):

$$\theta = \beta_0 + \beta_{\Delta t_{i:3}}$$

We call this the evoked response model.

Our model comparison logic was as follows. We first evaluated whether the posterior probability of the evoked response model was greater than that for the null model. This would suggest that time relative to a change point alone is a better predictor of decision dynamics than the average response. If the posterior probability of the evoked response model reliably exceeded the posterior probability of the null model, we then quantified the evidence for alternative models relative to the evoked response model. The sole effect of time relative to a change point was then framed as the new null hypothesis.

We used Bayes Factors to quantify the ratio of evidence for two competing hypotheses. If the ratio is close to 1, then the evidence is equivocal. As the ratio grows more positive, there is greater evidence for the model specified in the numerator, and if the ratio is less than 1, then there is evidence for the

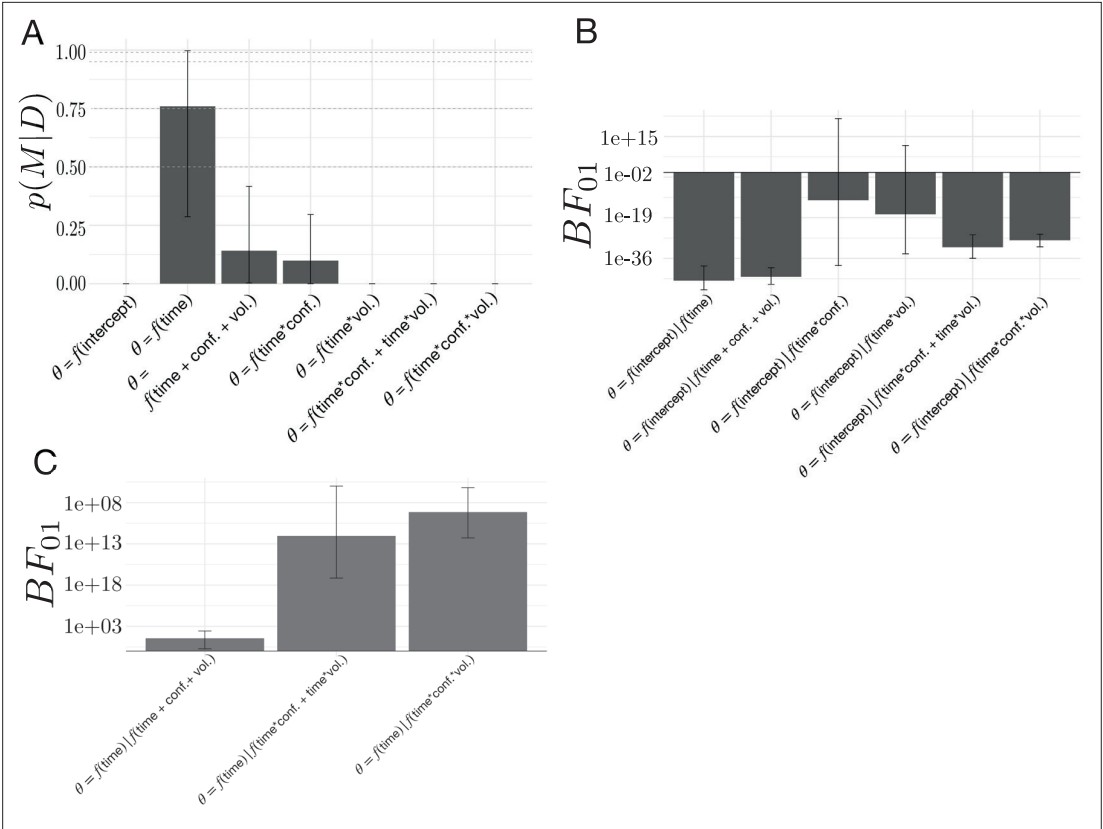

**Figure 8.** Model comparisons for the effect of volatility and conflict on the relationship between drift rate and boundary height. (**A**) The posterior probability for models testing for an effect of volatility and conflict on the angle of shift in $a$ and $v$, $\theta$. (**B**) The Bayes Factor for the null model relative to the alternative models specifying either an effect of time relative to a change point alone or a conditional effect on this evoked response $\theta$. (**C**) The Bayes Factor for the evoked response model relative to the surviving alternative models specifying a conditional effect on the evoked response, $\theta$. Note that time refers to time relative to the onset of a change point. All models specifying an interaction also include main effects. Dotted horizontal lines refer to grades of evidence (**Wagenmakers, 2007**).

model specified in the denominator (**Jeffreys, 1998**). Evidence for the null hypothesis is denoted $BF_{01}$ and evidence for the alternative hypothesis is denoted $BF_{10}$. Because Experiment 2 took a within-subject approach, a separate model was fit for each participant for all proposed models.

To determine whether volatility and conflict affected these peri-change decision dynamics, we modeled changes in decision policy on the drift rate and boundary height surface as a function of $\lambda$ and $p$, where $\lambda$ corresponds to the average period of stability and $p$ corresponds to the mean probability of reward for the optimal choice (see **Figure 8** for the full set of models considered). We explored the potential influence of volatility and conflict on the relationship between drift rate and boundary height by examining the posterior probability for each hypothesized model given the set of alternative hypotheses (Model proposals and evaluation; **Figure 8A**). We found that the evoked response model describing the relationship between shifts in decision parameters and time relative to a change point was more probable than the null model (see **Figure 8A**).

We also present the evidence for the null model against each alternative model as a Bayes Factor ($BF_{01}$) (**Figure 8B**). The 95% confidence interval for the $BF01$ comparing the ratio of evidence for the null model and the evoked response model specifying time-dependent effects of volatility included 1, suggesting inconclusive evidence for either of these models. Likewise, the 95% confidence interval for the $BF_{01}$ comparing the evidence for the null model against the model specifying change-point-evoked effects of conflict included 1, suggesting no substantive difference between them. Given the equivocal evidence for these two models we excluded them from further comparison with the evoked response model.

The remainder of the models had substantially negative $BF_{01}$ values (**Figure 8B**), suggesting that they better fit the data than the null model and allowing them to survive to the next stage of analysis.

To evaluate the hypothesis that time alone best accounted for the data, we computed the $BF_{01}$ for the evoked response model against the surviving models from the null model analysis. We find that, for all the remaining models, the $BF_{01}$ is substantially positive (*Figure 8C*), indicating that the evoked response model best accounted for the data (posterior probability of evoked response model given the set of models considered: $0.76 \pm 0.473$; posterior prob. for 3/4 participants > 0.99).

These analyses suggest that the relationship between the rate of evidence accumulation and the boundary height is only related to the change point itself. We find no evidence to suggest that changing the degree of volatility or changing the degree of conflict changes the path of the decision policy following a change point. Thus, the stereotyped response of the decision policy is solely dependent on the presence of a change point rather than either the history of change point frequency or the history of optimal choice ambiguity. Note that while the ideal observer estimates respond to our conditional manipulations of volatility and conflict, the decision dynamics $\theta$ we observe do not reflect these effects. This is due to the noisy, imperfect correspondence between the ideal observer signals and $a$ and $v$. This suggests that adaptation to environmental changes in action-outcome contingencies involves a rapid, coordinated increase in the relationship between the amount of information needed to make a decision and a decrease in the rate of information accumulation, with a stereotyped return to a stable baseline soon thereafter until another change occurs.

## No evidence for locus-coeruleus norepinephrine (LC-NE) system contribution to the decision trajectory

The LC-NE system is known to modulate exploration states under uncertainty and pupil diameter shows a tight correspondence with LC neuron firing rate (*Aston-Jones and Cohen, 2005*; *Rajkowski et al., 1994*), with changes in pupil diameter indexing the explore-exploit decision state (*Jepma and Nieuwenhuis, 2011*). Similar to the classic Yerkes-Dodson curve relating arousal to performance (*Yerkes and Dodson, 1908*), performance is optimal when tonic LC activity is moderate and phasic LC activity increases following a goal-related stimulus (*Aston-Jones et al., 1999*, but see *Joshi et al., 2016* for an exception). Because of this link between LC-NE and the regulation of behavioral variability in response to uncertainty, we expected that LC-NE system responses, as recorded by pupil diameter, would associate with environmental uncertainty and the trajectory through decision policy space following a change in action-contingencies. Specifically, if the LC-NE system were sensitive to a change in the optimal choice then we should observe a moderate spike in phasic activity following a change in action-outcome contingencies. Note that we do not observe previously established links

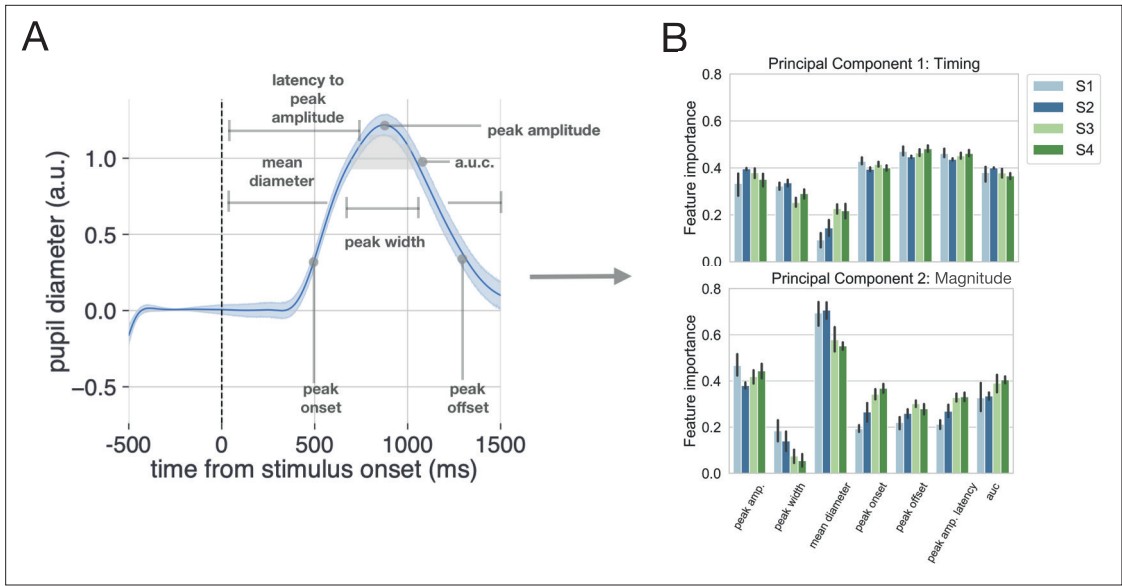

**Figure 9.** Method for analyzing pupil data. (**A**) The evoked pupillary response was characterized according to seven metrics. (**B**) These pupillary features were submitted to a principal component analysis. The contribution of each feature to the variance explained for the first two components is plotted for each subject. Note that we also conducted a supplementary analysis of the task-evoked pupillary response using a more conventional method with similar results.

between exploratory choice behavior and the pupillary response (*Jepma and Nieuwenhuis, 2011*; *Murphy et al., 2011*; *van Kempen et al., 2019*). We ask the reader to titrate their interpretation of these pupillary data accordingly.

We characterized the evoked pupillary response on each trial in Experiment 2 using seven metrics: the mean of the pupil data over each trial interval, the latency to the peak onset and offset, the latency to peak amplitude, the peak amplitude, and the area under the curve of the pupillary response (see Pupil data preprocessing; *Figure 9A*). From a computational perspective, reducing the dimensionality of this set of pupillary response metrics expands the set of models we can consider without taxing computational resources in a reasonable amount of time. Further, dimensionality reduction of the pupillary response allows us to capture separable sources of variance relating to timing and amplitude effects without restricting the data to a smaller set of metrics and possibly discarding information (e.g. timing effects may not be constrained to peak latency or onset latency; amplitude effects may not be constrained to peak dilation amplitude). Therefore, we submitted these metrics to principal component analysis to reduce their dimensionality while capturing maximum variance.

Evoked response characterization and principal component analyses were conducted for each session and for each subject in Experiment 2. The 95% CI for the number of principal components needed to explain 95% of the variance in the data was calculated over subjects and sessions to determine the number of principal components to keep for further analysis. To aid in interpreting subsequent analysis using the selected principal components, the feature importance of each pupil metric was calculated for each principal component and aggregated across subjects as a mean and bootstrapped 95% CI (*Figure 9*). We found that the first two principal components explained 95% of the variance in the pupillary data. Peak onset, peak offset, and latency to peak amplitude had the greatest feature importance for the first principal component (*Figure 9B*, upper panel). Mean pupil diameter and peak amplitude had the greatest feature importance for the second principal component (*Figure 9B*, lower panel). Thus, for interpretability, we refer to the first and second principal components as timing and magnitude components, respectively (*Figure 9B*). Note that we also conduct this analysis using more conventional methods of pupillary analysis and continue to observe a null effect (see Pupil data preprocessing for details).

To test for the possibility that fluctuations in norepinephrine covaried with changes in the drift-rate and the boundary height, we evaluated a set of models exploring the relationship between the timing and magnitude components of the change-point-evoked pupillary response and shifts in $\theta$.

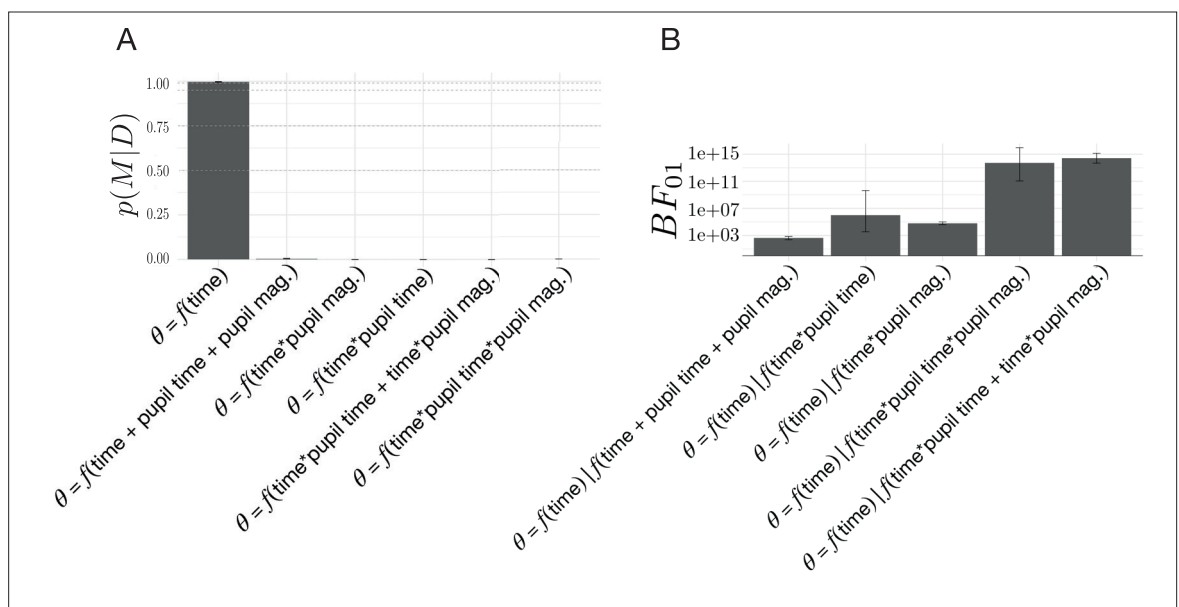

**Figure 10.** Model comparisons for the effect of change-point-evoked pupillary dynamics on the relationship between drift rate and boundary height ($\theta$). (**A**) The posterior probability for models testing for an effect of pupillary dynamics on $\theta$. (**B**) The Bayes Factor for the evoked response model relative to the alternative models specifying an effect of pupillary dynamics on the evoked response, $\theta$. Note that time refers to time relative to the onset of a change point. All models specifying an interaction also include main effects.

As in our previous model comparison (**Figure 8**; see Environmental instability prompts a stereotyped decision trajectory), we found that the model describing the relationship between decision policy shift and time relative to a change point had the highest posterior probability given the set of models considered (**Figure 10A**). To further evaluate the extent of the evidence for the evoked response hypothesis, we present the evidence for the evoked response model against the original model set as $BF_{01}$ (**Figure 10B**). We find unambiguous evidence in favor of the evoked response model relative to the models specifying the modulation of $\theta$ via the timing and magnitude features of the change-point-evoked pupillary response (posterior probability of time-null model given the set of models considered: $0.997 \pm 0.002$), with substantially positive $BF_{01}$ values. We find no evidence that the pupillary response associates with the dynamics of the decision policy changes in response to a change in action-outcome contingencies.

## Discussion

We investigated how decision policies change when the rules of the environment change. In two separate experiments, we characterized how decision processes adapted in response to a change in action-outcome contingencies as a trajectory through the space of possible types of exploratory and exploitative decision policies. Our findings highlight how, in the context of two choice paradigms, when faced with a possible change in outcomes, humans rapidly shift to a slow exploratory strategy by reducing the drift rate and, sometimes, increasing the boundary height in a stereotyped manner. Using pupillary data, we were unable to detect a relationship between the LC-NE system and the dynamics of adaptive decision policies in unstable environments. Our findings show how the underlying decision algorithm adapts to different forms of uncertainty.

Exploration and exploitation states are not discrete, but exist along a continuum (**Addicott et al., 2017**). Instead of switching between binary states, humans manage environmental instability by adjusting the greediness of their decision policies (**Sadeghiyeh et al., 2020**; **Prat-Carrabin et al., 2020**; **Feng et al., 2020**; **Wilson et al., 2014**; **Payzan-LeNestour and Bossaerts, 2011**; **Payzan-Lenestour and Bossaerts, 2012**; **Wilson et al., 2021**). Depending on the relative configuration of parameters in the accumulation to bound process, this adjustment can manifest as either speeded or slowed decisions (**Figure 1E**; **Alexandrowicz, 2020**; **Ratcliff, 1978**). Our results suggest that, in the context of volatile two-choice decisions, humans adopt a mechanism that simultaneously changes the rate of evidence accumulation and, sometimes, the threshold of evidence needed to trigger a decision, so as to adapt to an environmental change (**Figure 6A**). As soon as a shift in action outcomes is suspected, an internal estimate of change point probability increases and an estimate of the belief in the value of the optimal target plummets (**Figure 7A**). The rapid increase in change point probability causes a rapid *rise* in the boundary height on the subsequent trial, thereby increasing the criterion for selecting a new action and allowing variability in the accumulation process to have a greater influence on choice (**Figure 7B**), although this latter effect is inconsistent across experiments. These changes lead to slow exploratory decisions that facilitate discovery of the new optimal action and result in a quick recovery of the original threshold value over the course of a few trials. In parallel, the rate of evidence accumulation for the optimal choice decreases, with an immediate drop that gradually returns to its asymptotic value as the belief in the value of the optimal choice stabilizes. These results show that when a learner confronts a change point, the decision policy becomes more exploratory by simultaneously increasing the amount of evidence needed to make a decision *and* slowing the integration of evidence over time. Together, these decision dynamics form a mechanism for gathering information in an unbiased manner that slows the decision at the decision process level but responds quickly relative to a suspected change in trial time.

Critically, our finding that underlying decision policies can reconfigure multiple underlying decision parameters closely parallels recent work in the domain of information-seeking. Information seeking has been decomposed into random and directed components (**Wilson et al., 2014**). Random exploration refers to inherent behavioral variability that leads us to explore other options, while directed exploration refers to the volitional pursuit of new information. Feng and colleagues recently found that random exploration is driven by changes in the drift rate and the boundary height, with drift rate changes dominating the policy shift (**Feng et al., 2020**). When environmental conditions encouraged exploration, the drift rate slowed, reducing the signal-to-noise ratio of the reward representation. This finding clearly aligns with our current observations showing that the drift rate sharply decreases in

response to a change point and that this change in drift rate dominates the reconfiguration of decision processes, although our experiments were not designed to isolate the directed and random elements of exploration.

Our results are also broadly consistent with a growing body of research converging on the idea that decision policies are not static, but sensitive to changes in environmental dynamics (*Dunovan and Verstynen, 2019*; *Urai et al., 2018*). Previous work by our lab (*Dunovan and Verstynen, 2019*) has shown how, during a modified reactive inhibitory control task, different feedback signals target different parts of the accumulation-to-bound process. Specifically, errors in response timing drove rapid changes in the drift rate on subsequent trials, while selection errors (i.e. making a response on trials where the response should be inhibited) changed the boundary height. Further, there is new evidence that the drift rate adapts on the basis of previous choices, independent of the feedback given for those choices. Urai and colleagues have convincingly demonstrated that choice history signals sculpt the dynamics of the accumulation process by biasing the rate of evidence accumulation (*Urai et al., 2018*). Our current findings and these previous observations (*Pedersen et al., 2017*; *Ratcliff and Frank, 2012*) all highlight how sensitive the parameters of accumulation-to-bound processes are to immediate experience.

Previous literature has shown a conflict-induced spike in reaction time (e.g. *Jahfari et al., 2019*). However, our complex reaction time results depart from this. One reason for this departure may relate to the demands of the task we are asking participants to perform. While increased cognitive demand should increase reaction times across conditions, we observe a linear decrease in reaction time as a function of volatility when conflict is highest, and we also see that a net increase in conflict decreases reaction times (*Figure 3D*). We suspect that the presence of both conflict and volatility blurs the distinction between these two sources of uncertainty, especially under high volatility and high conflict conditions. We also see this effect in our formulation of change point probability (CPP), with a bias to overestimate CPP when conflict is high (*Figure 4D*). It is possible that participants also exhibit this bias to overestimate volatility when conflict is high, which could muddle the effect of conflict on reaction times. Future research should explore the interaction of change point and conflict estimation on the speed-accuracy trade-off.

We hypothesized that any shift in decision policy in response to a change in action-outcome contingencies would be linked to changes in phasic responses of the LC-NE pathways (*Aston-Jones and Cohen, 2005*). However, we failed to find any evidence of this link using pupillary responses as a proxy of LC-NE dynamics. It should be noted, however, that our experimental design cannot distinguish between pupillary dynamics driven by other catecholamines, such as dopamine, and those dynamics driven by the LC-NE system (*Spiers and Calne, 1969*; *McClure et al., 2005*; *Gershman and Tzovaras, 2018*; *Gershman and Uchida, 2019*), Thus it is possible that the LC-NE system may still be playing a role in shift of decision policies, and the pupil responses we collected were insensitive to the underlying dynamics. Nonetheless, this null association suggests that an alternative neural mechanism drives the adaptive changes that we observed behaviorally.

One possible alternative mechanism for resetting decision policies is is dopaminergic changes to the cortico-basal ganglia-thalamic (CBGT) pathways, or 'loops'. Both recent experimental (*Yartsev et al., 2018*; *Dunovan et al., 2015*) and theoretical (*Bogacz and Larsen, 2011*; *Caballero et al., 2018*; *Wei et al., 2015*) studies have pointed to the CBGT loops as being a crucial pathway for accumulating evidence during decision making, with the wiring architecture of these pathways ideal for implementing the sequential probability ratio test (*Bogacz and Gurney, 2007*; *Bogacz, 2007*), the statistically optimal algorithm for evidence accumulation decisions and the basis for the DDM itself (*Ratcliff, 1978*). Further, multiple lines of theoretical work have suggested that, within the CBGT pathways, the difference in direct pathway activity between action channels covaries with the rate of evidence accumulation for individual decisions (*Mikhael and Bogacz, 2016*; *Bariselli et al., 2019*; *Dunovan et al., 2019*; *Rubin et al., 2021*), while the indirect pathways are linked to control of the boundary height (*Wei et al., 2015*; *Herz et al., 2016*; *Bogacz, 2007*; *Ratcliff and Frank, 2012*). This suggests that changes in the direct and indirect pathways, both within and between representations of different actions, may regulate shifts in decision policies.

Critically, the CBGT pathways are a target of the dopaminergic signaling that drives reinforcement learning (*Schultz et al., 1992*), suggesting that changes in relative action-value should drive trial-by-trial changes in the drift rate. Indeed, previous work relating dopaminergic circuitry to decision

policy adaptation suggests that dopamine may play a critical role in modulating decision policies. Dopamine has substantial links to exploration (*Kakade and Dayan, 2002*) and recent pharmacological evidence suggests a role for dopaminergic regulation of exploration in humans (*Chakroun et al., 2020*). More explicitly, both directed and random exploration have been linked to variations in genes that affect dopamine levels in prefrontal cortex and striatum, respectively (*Gershman and Tzovaras, 2018*). Physiologically, previous work has found that a dopamine-controlled spike-timing-dependent plasticity rule alters the ratio of direct to indirect pathway efficacy in a simulated corticostriatal network (*Vich et al., 2020*), with overall indirect pathway activity (i.e. pre-decision firing rates) linked to the modulation of the boundary height in a DDM and the difference in direct pathway activation across action channels associating with changes in the drift rate (*Dunovan et al., 2019*; *Rubin et al., 2021*). Moreover, recent optogenetic work in mice suggests that activating the subthalamic nucleus, a key node in the indirect pathway, not only halts the motoric response but also interrupts cognitive processes related to action selection (*Heston et al., 2020*). Our current observations, combined with this previous work, suggests that the decision policy reconfiguration that we observe may associate with similar underlying corticostriatal dynamics, with belief-driven changes to drift rate varying with the difference in direct pathway firing rates across action channels (*Dunovan et al., 2019*), and change-point-probability-driven changes to the boundary height varying with overall indirect pathway activity (*Dunovan et al., 2019*; *Vich et al., 2020*). Future physiological studies should focus on validating this predicted relationship between decision policy reconfiguration and CBGT pathways.

The current study raises many more questions about the dynamics of adaptive decision policies than it answers. For example, we only sparsely sampled the space of possible states of value conflict and volatility. Future work would benefit from a more complete sampling of the conflict and volatility space. A psychophysical characterization of how decision states shift in response to varying forms of uncertainty will expose potential non-linear relationships between the decision policy and feedback uncertainty. Moreover, the decisions that we have modeled here are simple two choice decisions, constrained mostly by the normative form of the traditional DDM framework (*Ratcliff, 1978*). Scaling the complexity of the task will allow for a more complete assessment of how these relationships change with more complex decisions that better approximate the choices that we make outside the lab. This could be done by moving the cognitive model to frameworks that can fit processes for decisions involving more than two alternatives (e.g. *Tajima et al., 2019*). Finally, because our estimate of the relationship between our ideal observer estimates of uncertainty and human estimates of uncertainty were indirect, this work would benefit from online approximations of ideal observer estimates, as has been done previously (*Wilson et al., 2010*). Indeed, there can be substantive individual differences in the detection of of change points (*Wilson et al., 2010*). Thus, an approximation of how well the estimates of change point probability from our ideal observer correspond to estimates that human observers hold is needed. This approximation would validate the fidelity of the relationship between the ideal observer estimates of uncertainty and the decision parameters that we observed.

Together, our results suggest that when humans are forced to change their mind about the best action to take, the underlying decision policy adapts in a specific way. When a change in action-outcome contingency is suspected, the rate of evidence accumulation decreases and more evidence may briefly be required to commit to a response, allowing variability inherent to the decision process to play a greater role in response selection and resulting in a *slow* exploratory state. As the environment becomes stable, the system gradually adapts to an exploitative state. Importantly, we find no evidence that norepinephrine pathways associate with this response. This suggests that other pathways may be engaged in this adaptive reconfiguration of decision policies. These results reveal the multifaceted underlying decision processes that can adapt action selection policy under multiple forms of environmental uncertainty.

## Materials and methods
### Participants
Neurologically healthy adults were recruited from the local university population. All procedures were approved by the Carnegie Mellon University Institutional Review Board (Approval Code: 2018_00000195; Funding: Air Force Research Laboratory, Grant Office ID: 180119). All research

participants provided informed consent to participate in the study and consent to publish any research findings based on their provided data.

Twenty-four participants (19 female, 22 right-handed, 19–31 years old) were recruited for Experiment 1 and paid $20 at the end of four sessions. Four participants (two female, 4 right-handed, 21–28 years old) were recruited for Experiment 2 and paid $10 for each of nine sessions, in addition to a performance bonus.

Processed data and code are available within a Github repository for this publication (copy archived at swh:1:rev:0486705db0f004a5e1365759f5f5a391790771f8, *Bond, 2021*). Hypotheses were registered prior to the completion of data collection using the Open Science Framework (*Foster, MSLS and Deardorff, MLIS, 2017*).

## Stimuli and procedure

### Experiment 1

To begin the task, each participant read the following instructions:

"You're going on a treasure hunt! You will start with 600 coins in your treasure chest, and you'll be able to pay a coin to open either a purple or an orange box. When you open one of those boxes, you will get a certain number of coins, depending on the color of the box. However, opening the same box will not always give you the same number of coins, and each choice costs one coin. After making your choice, you will receive feedback about how much money you have. Your goal is to make as much money as possible. Press the green button when you're ready to continue. Choose the left box by pressing the left button with your left index finger and choose the right box by pressing the right button with your right index finger. Note that if you choose too slowly or too quickly, you won't earn any coins. Finally, remember to make your choice based on the color of the box. Press the green button when you're ready to begin the hunt!".

On each trial, participants chose between one of two 'mystery boxes' presented side-by-side on the computer screen (*Figure 2A*). Participants selected one of the two boxes by pressing either a left button (left box selection) or right button (right box selection) on a button box (Black Box ToolKit USB Response Pad, URP48). Reaction time (RT) was defined as the time elapsed from stimulus presentation to stimulus selection. Reaction time was constrained so that participants had to respond within 100 ms to 1000 ms from stimulus presentation. If participants responded too quickly, the trial was followed by a 5 s pause and they were informed that they were too fast and asked to slow down. If participants responded too slowly, they received a message saying that they were too slow, and were asked to choose quickly on the next trial. In both of these cases, participants did not receive any reward feedback or earn any points, and the trial was repeated so that 600 trials met these reaction time constraints. In order to avoid fatigue, a small break was given midway through each session (break time: 0.72 ± 1.42 m). Participants began each condition with 600 points and lost one point for each incorrect decision.

Feedback was given after each rewarded choice in the form of points drawn from the normal distribution $N(\mu = 3, \sigma = 1)$ and converted to an integer. If the choice was unrewarded, then participants received 0 points. These points were displayed above the selected mystery box for 0.9 s. To prevent stereotyped responses, the inter-trial interval was sampled from a uniform distribution with a lower limit of 250ms and an upper limit of 750ms ($U(250, 750)$). The relative left-right position of each target was pseudorandomized on each trial to prevent incidental learning based on the spatial position of either the mystery box or the responding hand.

To induce decision-conflict, the probability of reward for the optimal target ($P$) was manipulated across two conditions. We imposed a relatively low probability of reward for the high conflict condition ($P = 0.65$). Conversely, we imposed a relatively high probability of reward for the low conflict condition ($P = 0.85$). For all conditions, the probability of the low-value target was $1 - P$.

Along with these reward manipulations, we also introduced volatility in the action-outcome contingencies. After a prespecified number of trials, the identity of the optimal target switched periodically. The point at which the optimal target switched identities was termed a *change point*. Each period of mean contingency stability was defined as an epoch. Consequently, each session was composed of multiple change points and multiple epochs. Epoch lengths, in trials, were drawn from a Poisson distribution. The lambda parameter was held constant for both high conflict and low conflict conditions ($\lambda = 25$).

To manipulate volatility, epoch lengths were manipulated across two conditions. The high volatility condition drew epoch lengths from a Poisson distribution where $\lambda = 15$ and the low volatility condition drew epoch lengths from a distribution where $\lambda = 35$. In these conditions manipulating volatility, the probability of reward was held constant ($P = 0.75$).

Each participant was tested under four experimental conditions: high conflict, low conflict, high volatility, and low volatility. Each condition was completed in a unique experimental session and each session consisted of 600 trials. Each participant completed the entire experiment over two testing days. To eliminate the effect of timing and its correlates on reward learning (*Byrne et al., 2017*; *Murray et al., 2009*), the order of conditions was counterbalanced across participants.

### Experiment 2

Experiment 2 used male and female Greebles (*Gauthier and Tarr, 1997*) as selection targets (*Figure 2B*). Participants were first trained to discriminate between male and female Greebles to prevent errors in perceptual discrimination from interfering with selection on the basis of value. Using a two-alternative forced choice task, participants were presented with a male and female Greeble and asked to select the female, with the male and female Greeble identities resampled on each trial. Participants received binary feedback regarding their selection (correct or incorrect). This criterion task ended after participants reached 95% accuracy (mean number of trials to reach criterion: 31.29, standard deviation over means for subjects: 9.99).

After reaching perceptual discrimination criterion for each session, each participant was tested under nine reinforcement learning conditions composed of 400 trials each, generating 3600 trials per subject in total. Data were collected from four participants in accordance with a replication-based design, with each participant serving as a replication experiment. Participants completed these sessions across three weeks in randomized order. Each trial presented a male and female Greeble (*Gauthier and Tarr, 1997*), with the goal of selecting the sex identity of the Greeble that was most profitable (*Figure 2B*). Individual Greeble identities were resampled on each trial; thus, the task of the participant was to choose the sex identity rather than the individual identity of the Greeble which was most rewarding. Probabilistic reward feedback was given in the form of points drawn from the normal distribution $N(\mu = 3, \sigma = 1)$ and converted to an integer, as in Experiment 1. These points were displayed at the center of the screen. Participants began with 200 points and lost one point for each incorrect decision. To promote incentive compatibility (*Hurwicz, 1972*; *Ledyard, 1989*), participants earned a cent for every point earned. Reaction time was constrained such that participants were required to respond within 0.1 and 0.75 s from stimulus presentation. If participants responded in $\leq .1$ s, $\geq 0.75$ s, or failed to respond altogether, the point total turned red and decreased by five points. Each trial lasted 1.5 s and reward feedback for a given trial was displayed from the time of the participant's response to the end of the trial.

To manipulate change point probability, the sex identity of the most rewarding Greeble was switched probabilistically, with a change occurring every 10, 20, or 30 trials, on average. To manipulate the belief in the value of the optimal target, the probability of reward for the optimal target was manipulated, with $P$ set to 0.65, 0.75, or 0.85. Each session combined one value of $P$ with one level of change point probability, such that all combinations of change point frequency and reward probability were imposed across the nine sessions (*Figure 2C*). As in Experiment 1, the position of the high-value target was pseudo-randomized on each trial to prevent prepotent response selections on the basis of location.

Throughout the task, the head-stabilized diameter and gaze position of the left pupil were measured with an Eyelink 1,000 desktop mount at 1000 Hz. Participants viewed stimuli from within a custom-built booth designed to eliminate the influence of ambient sources of luminance. Because the extent of the pupillary response is known to be highly sensitive to a variety of influences (*Sirois and Brisson, 2014*), we established the dynamic range of the pupillary response for each session by exposing participants to a sinusoidal variation in luminance prior to the reward-learning task. During the reward-learning task, all stimuli were rendered isoluminant with the background of the display to further prevent luminance-related confounds of the task-evoked pupillary response. To obtain as clean a trial-evoked pupillary response as possible and minimize the overlap of the pupillary response between trials, the inter-trial interval was sampled from a truncated exponential distribution with a minimum of 4 s, a maximum of 16 s, and a rate parameter of 2. The eyetracker was calibrated and

the calibration was validated at the beginning of each session. See Pupil data preprocessing for pupil data preprocessing steps.

## Models and simulations

### Q-learning simulations

A simple, tabular q-learning agent (**Sutton and Barto, 1998**) was used to simulate action selection in contexts of varying degrees of conflict and volatility. On each trial, $t$, the agent chooses which of two actions to take according to the policy

$$\pi_t = \frac{\exp^{\beta * Q_t}}{\Sigma \beta * Q_t}. \tag{1}$$

Here, $\beta$ is the inverse temperature parameter, $1/\tau$, reflecting the greediness of the selection policy and $Q_t$ is the estimated state-action value vector on that trial. Higher values of $\beta$ reflect more exploitative decision policies.

After selection, a binary reward was returned. This was used to update the $Q$ table according using a simple update rule

$$Q_{t+1} = Q_t + \alpha(reward - Q_t), \tag{2}$$

where $\alpha$ is the learning rate for the model.

On each simulation an agent was initialized with a specific $\beta$ value, ranging from 0.1 to 3. On each run the agent completed 500 trials at a specific conflict and volatility level, according to the experimental procedures described in Stimuli and Procedure. The total returned reward was tallied after each run, which was repeated for 200 iterations to provide a stable estimate of return for each agent and condition. The agent was tested on a range of pairwise conflict ($P(optimal) = 0.55 - 0.90$) and volatility ($\lambda = 10 - 100$) conditions.

After all agents were tested on all conditions, the $\beta$ value for the agent that returned the greatest average reward across runs was identified as the optimal agent for that experimental condition.

### Drift diffusion model simulations

A normative drift-diffusion model (DDM) process (**Ratcliff, 1978**) was used to simulate the outcomes of agents with different drift rates and boundary heights. The DDM assumes that evidence is stochastically accumulated as the log-likelihood ratio of evidence for two competing decision outcomes. Evidence is tracked by a single decision variable $\theta$ until reaching one of two boundary heights, representing the evidence criterion for committing to a choice. The dynamics of $\theta$ is given by.

$$d\theta = vdt + \sigma dW \text{ for } t > tr;$$
$$\theta(t \leq tr) = z/a \tag{3}$$

where $v$ is the mean strength of the evidence and $\sigma$ is the standard deviation of a white noise process $W$, representing the degree of noise in the accumulation process. The choice and reaction time (RT) on each trial are determined by the first passage of $\theta$ through one of the two decision boundaries $\{a, 0\}$. In this formulation, $\theta$ remains fixed at a predefined starting point $z/a \in [0, 1]$ until time $tr$, resulting in an unbiased evidence accumulation process when $z = a/2$. In perceptual decision tasks, $v$ reflects the signal-to-noise ratio of the stimulus. However, in a value-based decision task, $v$ can be taken to reflect the difference between Q-values for the left and right actions. Thus, an increase (decrease) in $Q_L - Q_R$ from 0 would correspond to a proportional increase (decrease) in $v$, leading to more rapid and frequent terminations of $\theta$ at the upper (lower) boundary $a$ (0).

Using this DDM framework, we simulated a set of agents with different configurations of $a$ and $v$. Each agent completed 1,500 trials of a 'left' (upper bound) or 'right' (lower bound) choice task, with $tr = 0.26$ and $z = \frac{a}{2}$. The values for $a$ were sampled between 0.05 and 0.2 in intervals of 0.005. The values for $v$ were sampled from 0 to 0.3 in 0.005 intervals. At the end of each agent run, the probability of selecting the left target, $P(L)$, and the mean RT were recorded.

### Cognitive model

Our a priori hypothesis was that the drift rate ($v$) and the boundary height ($a$) should change on a trial-by-trial basis according to two estimates of uncertainty from an ideal observer (**Bond et al., 2018**).

We adapted the below ideal observer calculations from a previous study (*Vaghi et al., 2017*; for the original formulation of this reduced ideal observer model and its derivation, see *Nassar et al., 2010*).

First, we assumed that reward feedback drove the belief in the reward associated with an action. We called the belief in the reward attributable to a given action $B$. This reward belief is learned separately for each action target. Given the chosen target ($c$) and the unchosen target ($u$), the belief in the mean reward for the chosen and unchosen targets on the next trial (trial $t + 1$) was calculated as:

$$
\begin{aligned}
B_{t+1,c} &= B_{t,c} + \alpha_t \delta_t, \\
B_{t+1,u} &= B_{t,u}(1 - \Omega_t) + \Omega_t E(r),
\end{aligned}
\tag{4}
$$

where $\alpha_t$ denotes the learning rate, $\delta_t$ the prediction error, and $\Omega_t$ the change point probability on the current trial $t$, as discussed below. $E(r)$ refers to the pooled expected value of both targets:

$$
E(r) = \frac{\bar{r}_{t0} + \bar{r}_{t1}}{2},
\tag{5}
$$

with $\bar{r}_{t0}, \bar{r}_{t1}$ fixed based on the imposed target reward probabilities.

The prediction error, $\delta_t$, was the difference between the reward obtained for the target chosen and the model belief:

$$
\delta_t = r_t - B_{t,c}.
\tag{6}
$$

The signed belief in the reward difference between optimal and suboptimal targets ($\Delta B$) was calculated as the difference in reward value belief between target identities:

$$
\Delta B_{t+1} = B_{t,opt} - B_{t,subopt}.
\tag{7}
$$

Model confidence ($\phi$) was defined as a function of change point probability ($\Omega$) and the variance of the generative distribution of points ($\sigma_n^2$), both of which formed an estimate of relative uncertainty ($RU$):

$$
RU_t = \frac{\Omega_t \sigma_n^2 + (1-\Omega_t)(1-\phi_t)\sigma_n^2 + \Omega_t(1-\Omega_t)(\delta_t \phi_t)^2}{\Omega_t \sigma_n^2 + (1-\Omega_t)(1-\phi_t)\sigma_n^2 + \Omega_t(1-\Omega_t)(\delta_t \phi_t)^2 + \sigma_n^2}.
\tag{8}
$$

Thus $\phi$ is calculated as:

$$
\phi_{t+1} = 1 - RU_t.
\tag{9}
$$

An estimate of the variance of the reward distribution, $\sigma_t^2$, was calculated as:

$$
\sigma_t^2 = \sigma_n^2 + \frac{(1-\phi_t)\sigma_n^2}{\phi_t}
\tag{10}
$$

where $\sigma_n$ is the fixed variance of the generative reward distribution.

The learning rate of the model ($\alpha$) was determined by the change point probability ($\Omega$) and the model confidence ($\phi$). Here, the learning rate was high if either (1) a change in the mean of the distribution of the difference in expected values was likely ($\Omega$ is high) or (2) the estimate of the mean was highly imprecise ($\sigma_t^2$ was high):

$$
\alpha_t = \Omega_t + (1 - \Omega_t)(1 - \phi_t).
\tag{11}
$$

To model how learners update action-values, we calculated an estimate of how often the same action gave a different reward (*Vaghi et al., 2017*). This estimate gave our representation of change point probability, $\Omega$. The change point probability approached one from below as the probability of a sample coming from a uniform distribution, relative to a Gaussian distribution, increased:

$$
\Omega_t = \frac{U(r_t)H}{U(r_t)H + N(r_t | B_{\Delta_t}, \sigma_t^2)(1-H)}.
\tag{12}
$$

In *equation (12)*, $H$ refers to the hazard rate, or the global probability of a change point over trials:

$$
H = \frac{n_{cp}}{n_{trials}}.
\tag{13}
$$

Our preregistered expectation was that the belief in the value of a given action and an estimate of environmental stability would target different parameters of the DDM model. Specifically, we hypothesized that the belief in the relative reward for the two choices, $\Delta B$, would update the drift rate, $v$, or the rate of evidence accumulation:

$$v_{t+1} = \hat{\beta}_v \cdot \Delta B_t + v_t \tag{14}$$

while the change point probability, $\Omega$, would increase the boundary height, $a$, or the amount of evidence needed to make a decision:

$$a_{t+1} = a_0 + \hat{\beta}_a \cdot \Omega_t. \tag{15}$$

## Hierarchical drift diffusion modeling

First, to identify which decision parameters were sensitive to the onset of a change point, we estimated the posterior distribution of drift rate ($v$), boundary height ($a$), drift criterion ($dc$), starting point ($z$), and non-decision time ($t$) for the trial preceding the change point and the following three trials using stimulus-coded fitting methods for Experiment 1. We then looked for change-point-evoked effects in these parameters by comparing the overlap of the distributions for each decision parameter for each of these trials. If less than 5% of the mass of the trial-wise posterior distributions for a given decision parameter overlapped, we considered those distributions to exhibit change point sensitivity.

To identify the fits that best accounted for the data, we conducted a model selection process using Deviance Information Criterion (DIC) scores. We compared the set of fitted models (*Table 1*) to an intercept-only regression model ($DIC_i - DIC_{intercept}$). A lower DIC score indicates a model that loses less information. Here, a difference of $\leq$ two points from the lowest-scoring model cannot rule out the higher scoring model; a difference of 3–7 points suggests that the higher scoring model has considerably less support; and a difference of 10 points suggests essentially no support for the higher scoring model (*Spiegelhalter et al., 2002*; *Burnham and Anderson, 1998*).

We used these complementary model 'pruning' methods (i.e. distributional overlap and information loss) as an out-of-set filtering method to determine which decision parameters to include for the subsequent HDDM regression analyses in Experiment 2.

The best parameter fits, evaluated as above, were used to plot the decision trajectory (Decision vector representation) and to estimate the change-point-evoked relationship between those winning parameters (Model proposals and evaluation).

For Experiment 2, to assess whether and how much the ideal observer estimates of change point probability ($\Omega$) and the belief in the value of the optimal target ($\Delta B$) updated the rate of evidence accumulation ($v$) and the amount of evidence needed to make a decision ($a$), we regressed the change-point-evoked ideal observer estimates onto the decision parameters using hierarchical drift diffusion model (HDDM) regression (*Wiecki et al., 2013*). These ideal observer estimates of environmental uncertainty served as a more direct and continuous measure of the uncertainty we sought to induce with our experimental conditions (see *Figure 4* for how the experimental conditions impacted these estimates). Considering this more direct approach, we pooled change point probability and belief across all conditions and used these values as our predictors of drift rate and boundary height. Responses were accuracy-coded, and the belief in the difference between targets values was transformed to the belief in the value of the optimal target ($\Delta B_{\text{optimal(t)}} = B_{\text{optimal(t)}} - B_{\text{suboptimal(t)}}$). This approach allowed us to estimate trial-by-trial covariation between the ideal observer estimates and the decision parameters relative to the onset of a change point.

For both the HDDM fits for Experiment 1 and the regression analyses for Experiment 2, Markov-chain Monte-Carlo methods were used to sample the posterior distributions of the regression coefficients. Twenty thousand samples were drawn from the posterior distributions of the coefficients for each model, with 5000 burned samples and a thinning factor of five. We chose this number of samples to optimize the trade-off between computation time and the precision of parameter estimates, and all model parameters converged to stability. This method generates a distributional estimate of the regression coefficients instead of a single best fit.

To test our hypotheses regarding these HDDM regression estimates, we again used the posterior distributions of the regression parameters. To quantify the reliability of each regression coefficient, we computed the probability of the regression coefficient being greater than or less than 0 over the

posterior distribution. We considered a regression coefficient to be reliable if the estimated coefficient maintained the same sign over at least 95% of the mass of the posterior distribution.

## Analyses

### General statistical analysis

Statistical analyses and data visualization were conducted using custom scripts written in R (R Foundation for Statistical Computing, version 3.4.3) and Python (Python Software Foundation, version 3.5.5).

To determine how many trials would be needed to detect proposed condition effects, we conducted a power analysis by way of parameter recovery. For this, we simulated accuracy and reaction time data using our hypothesized model (Cognitive model) and calculated the generative or "true" mean drift rate and boundary height parameters across trials. Then we conducted hierarchical parameter estimation given 200, 400, 600, 800, or 1000 simulated trials. The mean squared error of parameter estimates was stable at 600 trials for all decision parameters. Additionally, as a validation measure, we estimated parameters using component models (drift rate alone, boundary height alone) and a combined model (drift rate and boundary height). We found that the Deviance Information Criterion (DIC) scores among competing models were clearly separable at 600 trials, and in favor of the hypothesized model from which we generated the data, as expected (Acknowledgments). Based on these results, we used 600 trials per condition for each participant for our first experiment. We chose to recruit 24 participants for this experiment to fully counterbalance the four conditions (4! = 24).

Binary accuracy data were submitted to a mixed effects logistic regression analysis with either the degree of conflict (the probability of reward for the optimal target) or the degree of volatility (mean change point frequency) as predictors. The resulting log-likelihood estimates were transformed to likelihood for interpretability. RT data were log-transformed and submitted to a mixed effects linear regression analysis with the same predictors as in the previous analysis. To determine if participants used ideal observer estimates to update their behavior, two more mixed effects regression analyses were performed. Estimates of change point probability and the belief in the value of the optimal target served as predictors of reaction time and accuracy across groups. As before, we used a mixed logistic regression for accuracy data and a mixed linear regression for reaction time data.

Because we adopted a within-subjects design, all regression analyses of behavior modeled the non-independence of the data as constantly correlated data within participants (random intercepts). Unless otherwise specified, we report bootstrapped 95% confidence intervals for behavioral regression estimates. To prevent any bias in the regression estimates emerging from collinearity between predictors and to aid easy interpretation, all predictors for these regressions were mean-centered and standardized prior to analysis. The Satterthwaite approximation was used to estimate p-values for mixed effects models (*Satterthwaite, 1946*; *Luke, 2017*).

### Decision vector representation

To concisely capture the change-point-driven response in the relationship between the boundary height and the drift rate over time, we represented the relationship between these two decision variables in vector space. Trial-by-trial estimates of drift rate and boundary height were calculated from the winning HDDM regression equation and z-scored. Then the difference between each sequential set of $(a, v)$ coordinates was calculated to produce a vector length. The arctangent between these subtracted values was computed to yield an angle in radians between sequential decision vectors (*Figure 7B*).

For Experiment 1, these computations were performed from the trial prior to the onset of the change point to eight trials after the change point. The initial window of nine trials was selected to maximize the overlap of stable data between high and low volatility conditions (see Supp. Fig. References). This resulted in a sequence of angles formed between trials –1 and 0 ($\Delta t_1$ yielding $\theta_1$), 0 and 1 ($\Delta t_2$ yielding $\theta_2$), and so on. To observe the timescale of these dynamics, a circular regression (*Mulder and Klugkist, 2017*) was performed to determine how $\theta$ changed as a function of the number of trials after the change point:

$$\theta = \hat{\beta}_0 + \hat{\beta}_{\Delta_t} + \cdots \hat{\beta}_{\Delta_{t8}}.$$

To quantitatively assess the number of trials needed for $\theta$ to stabilize, we calculated the probability that the posterior distributions of the regression estimates (*Appendix 1—figure 6*) for sequential pairs of trials had equal means ($\theta_{\Delta_t} = \theta_{\Delta_{t+1}}$). This result (*Appendix 1—figure 7*) provided an out-of-set constraint on the timescale of the decision response to consider for analogous analyses in Experiment 2.

Experiment 2 used the stability convergence analysis from Experiment 1 to guide the timescale of further circular analyses and, thus, placed a constraint on the complexity of the models proposed (Model proposals and evaluation). Because Experiment 2 took a replication-based approach, a separate model was fit for each participant for all proposed models. We report the mean and 95% CI of the posterior distributions of regression parameter estimates and the mean and standard deviation of estimates across subjects.

The circular regression analyses used Markov-chain Monte-Carlo (MCMC) methods to sample the posterior distributions of the regression coefficients. For both experiments, 10,000 effective samples were drawn from the posterior distributions of the coefficients for each model (*Kruschke and Vanpaemel, 2015*). Traces were plotted against MCMC iteration for a visual assessment of equilibrium, the autocorrelation function was calculated to verify independence of MCMC steps, trace distributions were visually evaluated for normality, and point estimates of the mean value were verified to be contained within the 95% credible interval of the posterior distribution for the estimated coefficients.

## Pupil data preprocessing

Pupil diameter data were segmented to capture the interval from 500ms prior to trial onset to the end of the 1500ms trial, for a total of 2000ms of data per trial. While the latency in the phasic component of the task-evoked pupillary response ranges from 100 to 200ms on average (*Beatty, 1982*), suggesting that our segmentation should end 200ms after the trial ending, participants tended to blink after the offset of the stimulus and during the intertrial interval (see *Appendix 1—figure 8* for a representative sample of blink timing). Because of this, we ended the analysis window with the offset of the stimulus. Following segmentation, pupil diameter samples marked as blinks by the Eyelink 1000 default blink detection algorithm and zero- or negative-valued samples were replaced by linearly interpolating between adjacent valid samples. Pupil diameter samples with values exceeding three standard deviations of the mean value for that session were likewise removed and interpolated. Interpolated data were bandpass filtered using a 0.01–5 Hz second-order Butterworth filter. Median pupil diameter calculated over the 500ms prior to the onset of the stimulus was subtracted from the trial data. Finally, processed data were z-scored by session.

For each trial interval, we characterized the evoked response as the mean of the pupil data over that interval, the latency to peak onset and offset, the latency to peak amplitude, the peak amplitude, and the area under the curve of the phasic pupillary response (*Figure 9A*). We then submitted these metrics to principal component analysis to reduce their dimensionality while capturing maximum variance. Evoked response characterization and principal component analysis were conducted for each session and for each subject.

The 95% CI for the number of principal components needed to explain 95% of the variance in the data was calculated over subjects and sessions to determine the number of principal components to keep for further analysis.

To aid in interpreting further analysis using the selected principal components, the feature importance of each pupil metric was calculated for each principal component and aggregated across subjects as a mean and bootstrapped 95% CI (*Figure 9B*).

Note that we also conducted a similar analysis using more conventional methods to assess the task-evoked pupillary response and observed another null effect. Specifically, if we take the derivative of the evoked pupillary response with respect to time (*Reimer et al., 2016*) and then characterize the pupillary response with the above metrics and conduct principal component analysis, we again see no evidence for a relationship between the pupillary response and the decision trajectory. Additionally, we observe no relationship between our experimental manipulations of conflict and volatility and these metrics, or a change-point evoked shift in pre-stimulus pupillary response (*Gilzenrat et al., 2010*, *Appendix 1—figure 13*). As such, we caution the reader to view our pupillary results in light of this lack of replication of pre-established exploration-driven pupillary responses.

## Model proposals and evaluation

To assess the hypothesized influences on $\theta$ in Experiment 2, we began our model set proposal with a null hypothesis. Our null model estimates decision dynamics as a function of the intercept, or the average of $\theta$:

$$\theta = \beta_0.$$

Next, we estimated decision dynamics solely as a function of time relative to a change point, with the timescale of consideration determined by the results of the stability convergence analysis from Experiment 1. We call this the evoked response model:

$$\theta = \beta_0 + \beta_{\Delta_t}...\beta_{\Delta_{tn}}.$$

We first evaluated whether the posterior probability of the evoked response model given the data was greater than the posterior probability for the absolute null model. If the lower bound of the 95% CI of the posterior probability for the time-null model exceeded the upper bound of the 95% CI for the absolute null model (i.e the posterior probability was greater for the evoked response model and the CIs were non-overlapping), we proceeded to evaluate the evidence for alternative models relative to this evoked response model. We evaluated the statistical reliability of the posterior probabilities using a bootstrapped 95% CI computed over subjects.

We considered an explicit set of hypotheses regarding the effect of the change-point-evoked pupillary response on boundary height and drift rate dynamics (see *Figure 10* for the full set of models considered). The first two principal components of the set of pupil metrics, which we term the timing and magnitude components, respectively, were included in this model set to evaluate the effect of the timing and magnitude of noradrenergic dynamics on the change-point-evoked decision manifold. Under the assumption of a neuromodulatory effect on decision dynamics, these principal components were shifted forward by one trial to match the expected timing of the response to neuromodulation.

To determine whether perturbations of volatility and conflict affected change-point-evoked decision dynamics, we estimated the evoked decision dynamics as a function of $\lambda$ and $p$, where $\lambda$ corresponds to the average length of an epoch and $p$ corresponds to the mean probability of reward for optimal target selection (see *Table 1* for the full set of models considered).

We used Bayes Factors to quantify the ratio of evidence for competing hypotheses (*Wagenmakers, 2007*). To estimate whether these models accounted for decision dynamics beyond the effect of time relative to a change point alone, we calculate the Bayes Factor for the evoked response model relative to each candidate model ($BF_{01}$). Finally, we calculate the posterior probability of the null model given the full set of alternative models (*Wagenmakers, 2007*). Note that this approach assumes that each model has equal a priori plausibility.

Bayes Factor visualizations represent the mean and bootstrapped 95% CI with 1000 bootstrap iterations.

## Acknowledgements

We thank all members of the Cognitive Axon Lab for their feedback during the development of this work. We also thank Marlene Behrmann and Michael Granovetter for their help with eye-tracking and pupillometry data collection and Chris Wordingham for his programming and engineering consultation in the early phases of this project.

## Additional information

### Funding

| Funder | Grant reference number | Author |
| --- | --- | --- |
| Air Force Research Laboratory | FA9550-18-1-0251 | Krista Bond Timothy Verstynen |

The funders had no role in study design, data collection and interpretation, or the decision to submit the work for publication.

### Author contributions

Krista Bond, Conceptualization, Data curation, Formal analysis, Investigation, Methodology, Project administration, Resources, Software, Visualization, Writing - original draft, Writing - review and editing;

Kyle Dunovan, Conceptualization, Methodology, Software, Writing - review and editing; Alexis Porter, Data curation, Investigation, Project administration; Jonathan E Rubin, Conceptualization, Writing - review and editing; Timothy Verstynen, Conceptualization, Formal analysis, Funding acquisition, Investigation, Project administration, Resources, Supervision, Validation, Writing - review and editing

### Author ORCIDs
Krista Bond ⓘ http://orcid.org/0000-0003-1492-6798
Kyle Dunovan ⓘ http://orcid.org/0000-0002-7857-5133
Jonathan E Rubin ⓘ http://orcid.org/0000-0002-1513-1551
Timothy Verstynen ⓘ http://orcid.org/0000-0003-4720-0336

### Ethics
Human subjects: Neurologically healthy adults were recruited from the local university population. All procedures were approved by the Carnegie Mellon University Institutional Review Board (Approval Code: 2018_00000195; Funding: Air Force Research Laboratory, Grant Office ID: 180119). All research participants provided informed consent to participate in the study and consent to publish any research findings based on their provided data.

### Decision letter and Author response
Decision letter https://doi.org/10.7554/eLife.65540.sa1
Author response https://doi.org/10.7554/eLife.65540.sa2

## Additional files

### Supplementary files
• Transparent reporting form

### Data availability
Behavioral data and their computational derivatives are available at https://github.com/kmbond/dynamic_decision_policy_reconfiguration (copy archived at swh:1:rev:0486705db0f004a5e1365759f5f5a391790771f8). Code used to generate figures can be found here. Raw pupillometry data (DOI: 10.1184/R1/13543133), the features of the task-evoked pupillometry response (DOI: 10.1184/R1/13543067.v1), and the principal components calculated from those features (DOI: 10.1184/R1/13543160.v1) are available here.

The following dataset was generated:

| Author(s) | Year | Dataset title | Dataset URL | Database and Identifier |
|---|---|---|---|---|
| Bond K | 2021 | Raw pupillometry data | https://doi.org/10.1184/R1/13543133 | KiltHub, 10.1184/R1/13543133 |
| Bond K | 2021 | Features of task-evoked pupillary response | https://doi.org/10.1184/R1/13543067.v1 | KiltHub, 10.1184/R1/13543067.v1 |
| Bond K | 2021 | Principal components of task-evoked pupillary response | https://doi.org/10.1184/R1/13543160.v1 | KiltHub, 10.1184/R1/13543160.v1 |

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

## Appendix 1

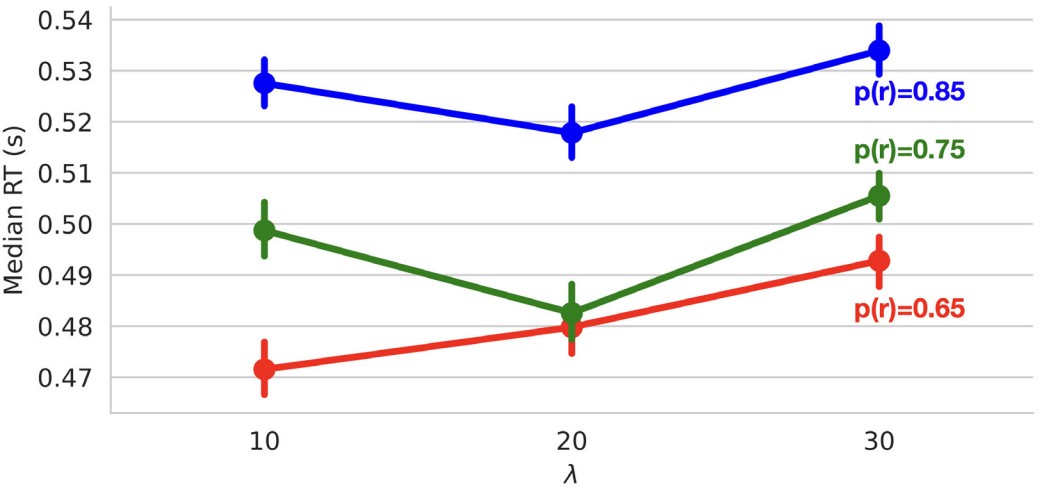

**Appendix 1—figure 1.** Reaction times. Median reaction times as a function of volatility (epoch length; $\lambda$) and conflict (probability of reward; $p(r)$).

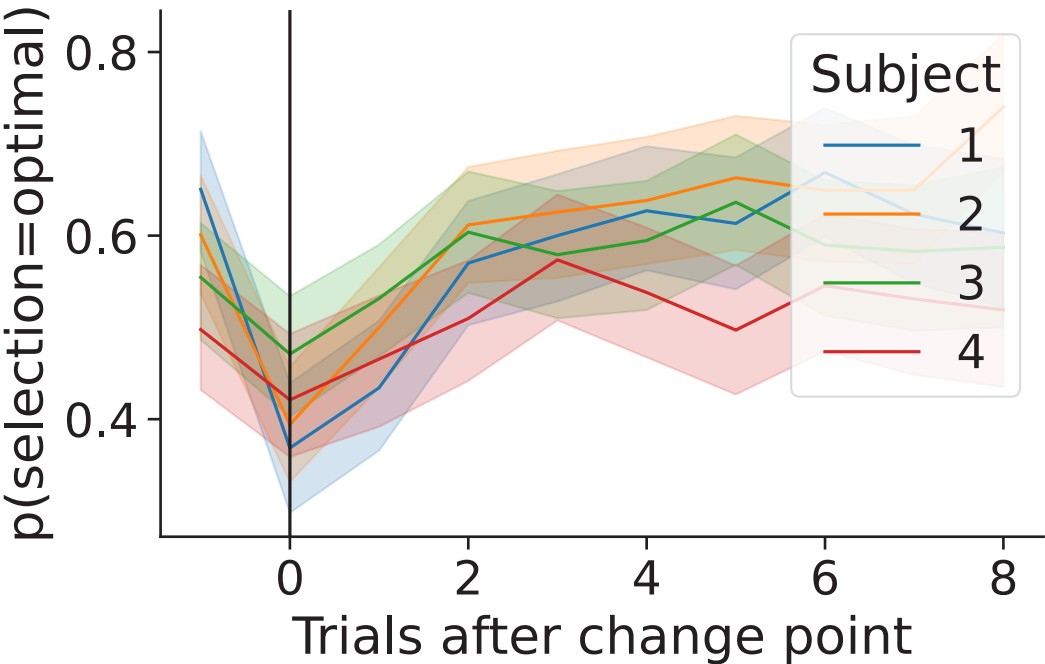

**Appendix 1—figure 2.** Change-point-evoked accuracy. Change-point-evoked accuracy by subject.

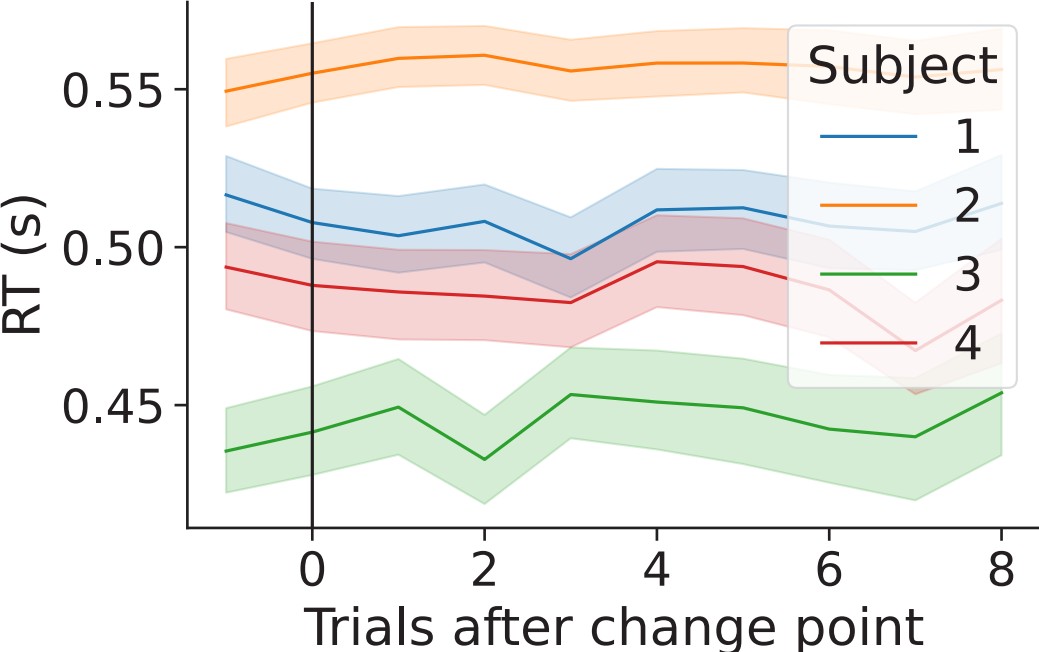

**Appendix 1—figure 3.** Change-point evoked reaction times. Change-point-evoked reaction times by subject.

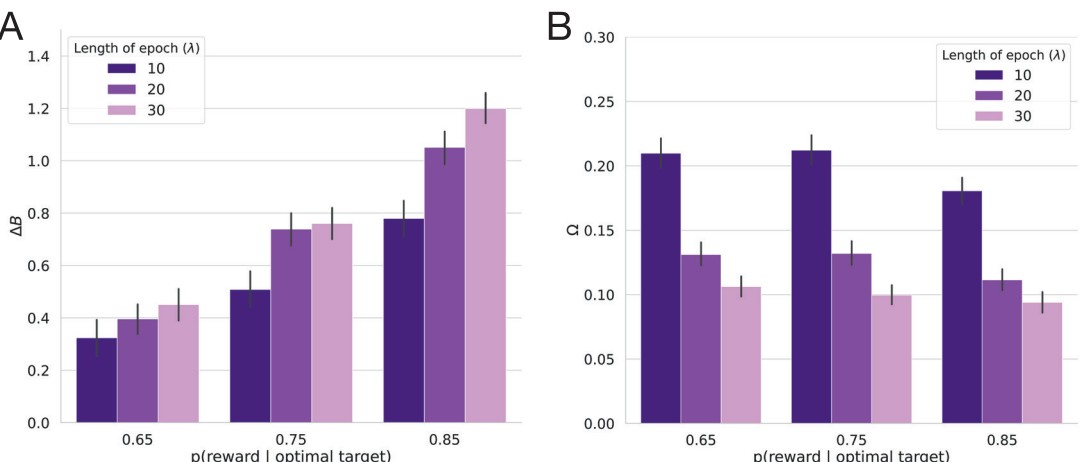

**Appendix 1—figure 4.** Ideal observer estimates for Experiment 2. (**A**) The average belief in the value of the optimal target ($\Delta B$) as a function of the probability of reward (conflict) and the average period of stability for the optimal choice ($\lambda$; volatility). (**B**) Average change point probability ($\Omega$) as a function of conflict and volatility.

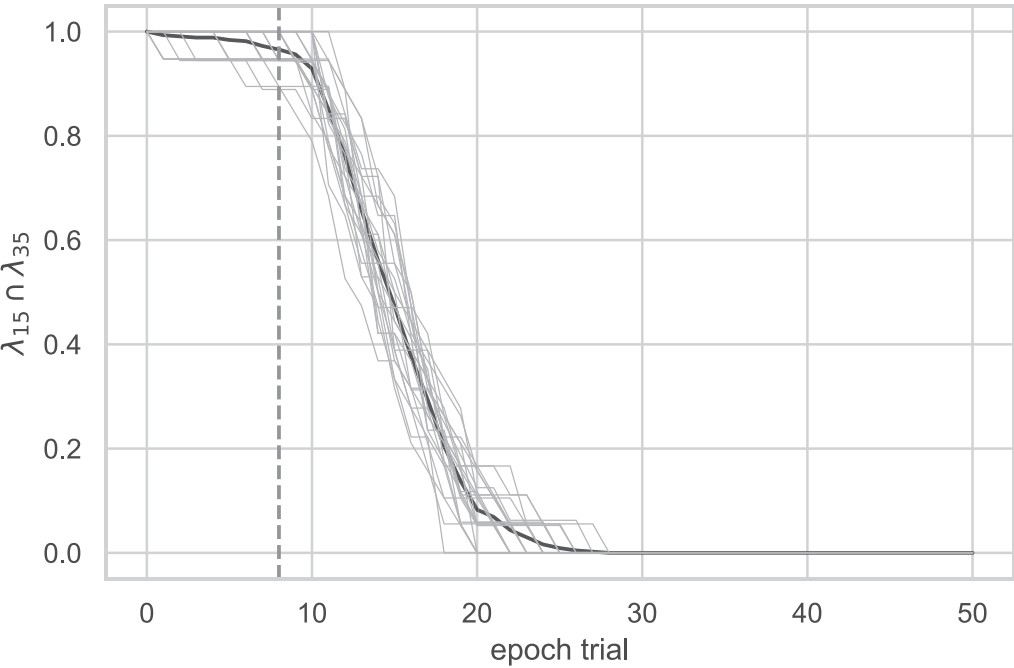

**Appendix 1—figure 5.** Initial window selection. Analysis conducted on data from Experiment 1 to determine the timescale of the response that maximized the intersection between high volatility ($\lambda = 15$) and low volatility ($\lambda = 35$) data. The bolded line represents the mean and the gray lines represent individual subjects. The dotted line indicates the initial window of nine trials used.

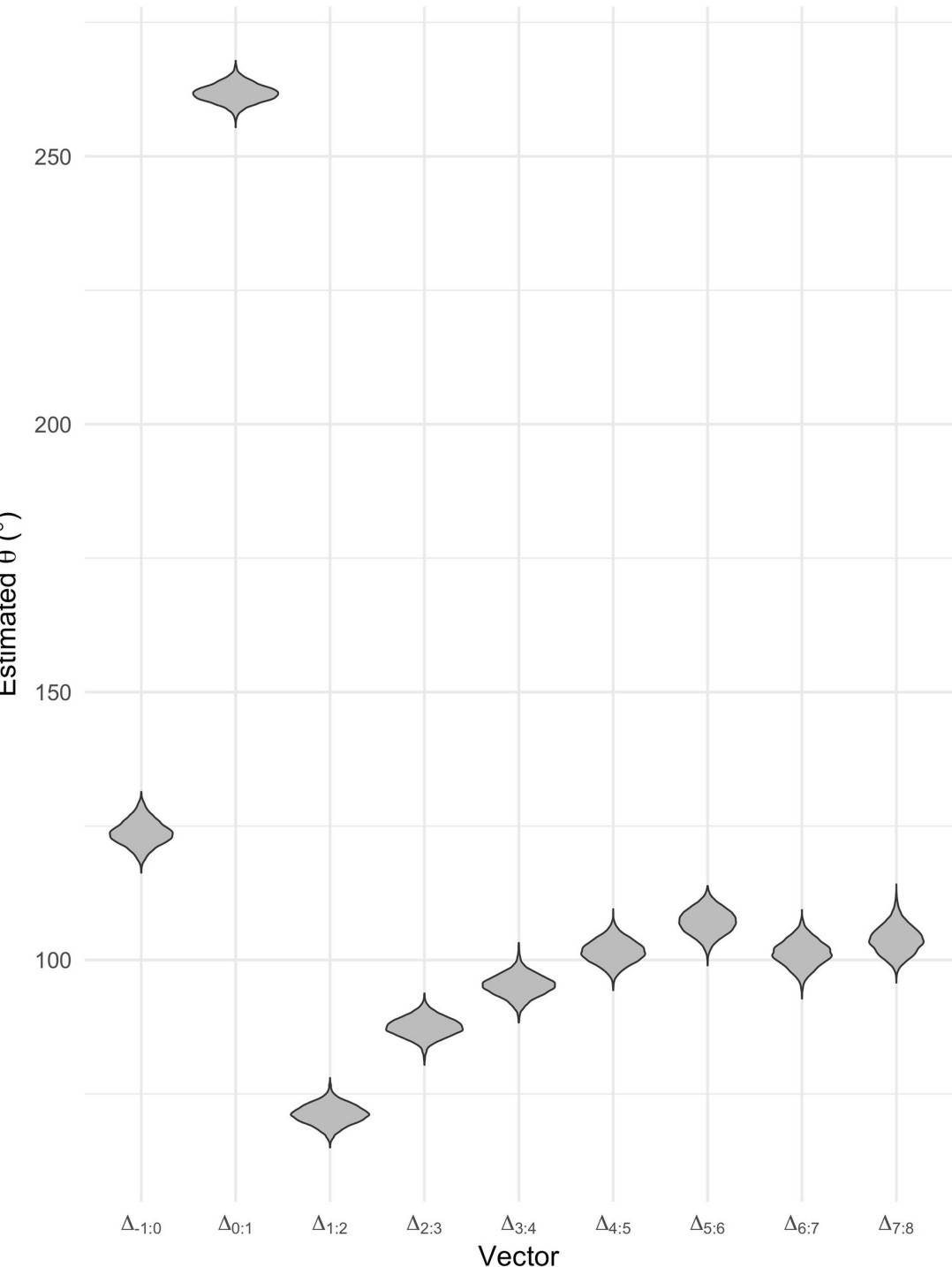

**Appendix 1—figure 6.** Stability analysis. Analysis conducted on Experiment 1 to determine the timescale of the response to consider for Experiment 2. The estimated angle is plotted as a function of time within an epoch (estimate from circular regression).

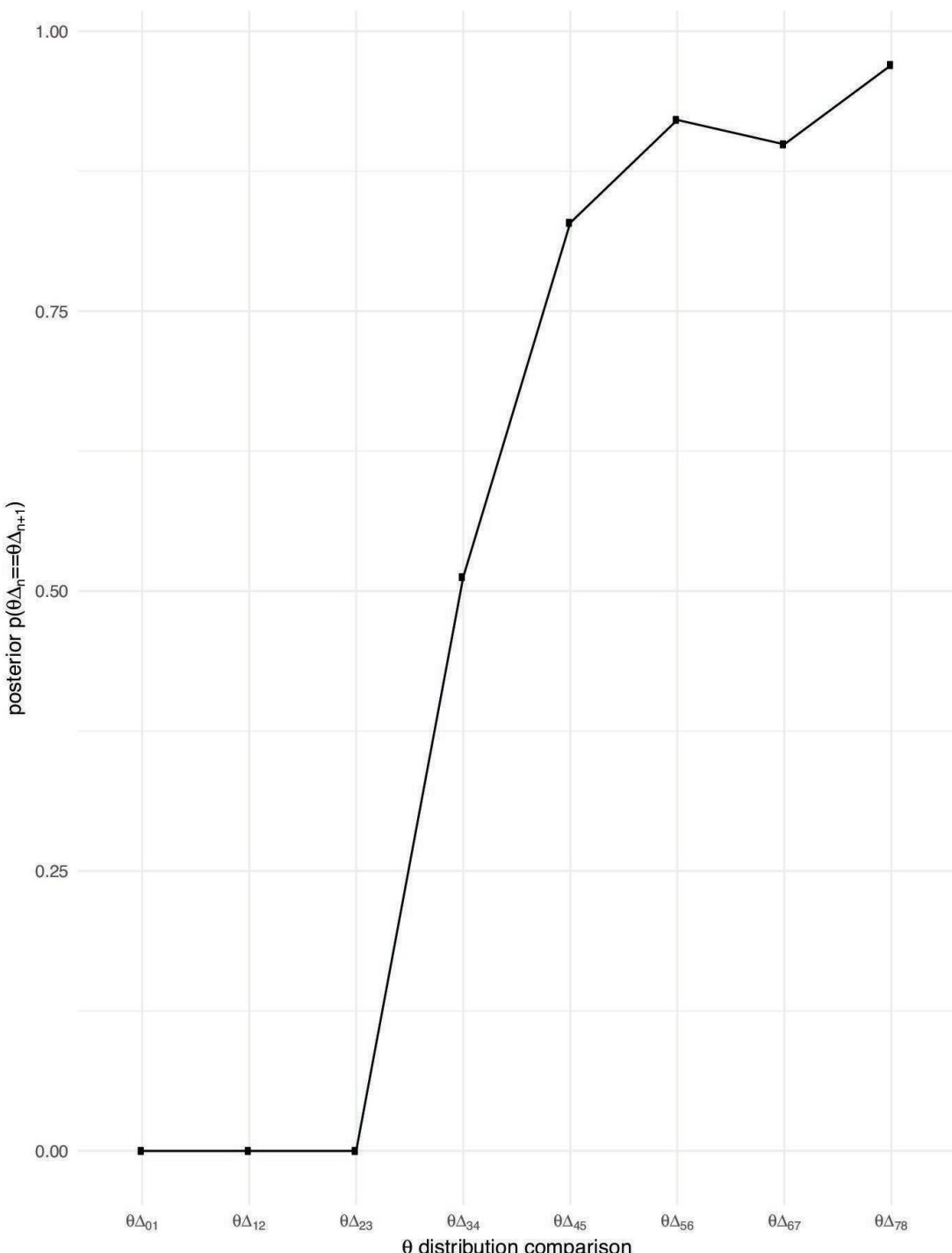

**Appendix 1—figure 7.** Quantification of stability. Probability that sequential posterior distributions for $\beta_{\Delta t}$ have equal means.

tepr_sub-789_sess-5_cond-6510_random_raster

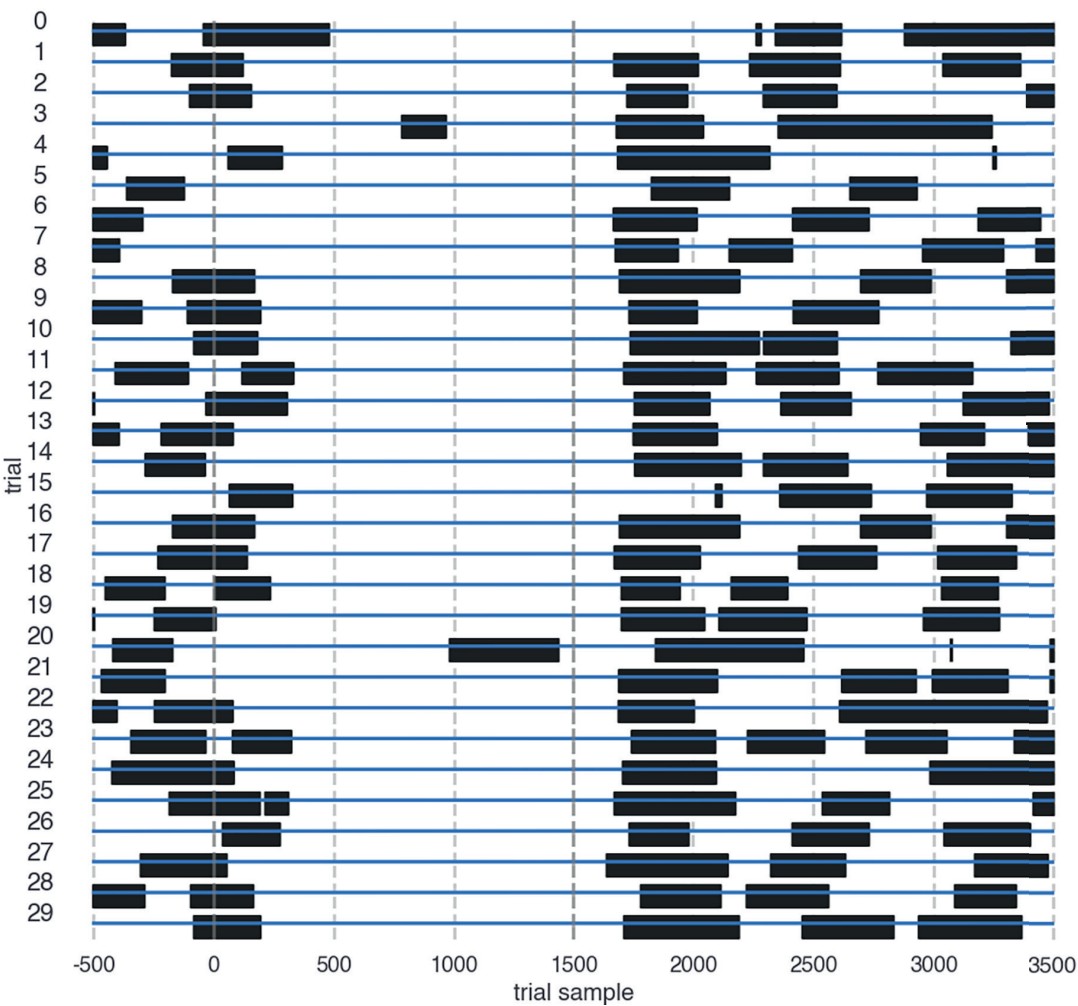

**Appendix 1—figure 8.** Blink timing. Blink timing for a sample participant. For visibility, thirty trials were selected at random. The onset of the trial is marked as time 0 and the trial ends at 1500ms. Blinks are marked in black. Blink timing plots are available for all subjects and all conditions in the GitHub repository for this publication.

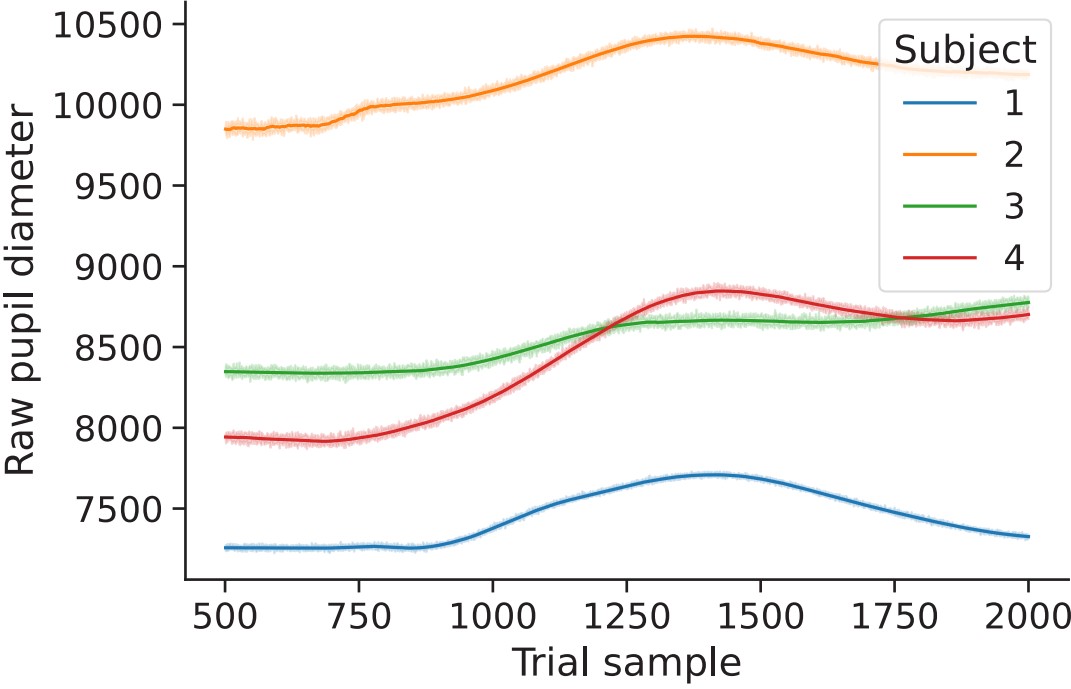

**Appendix 1—figure 9.** Raw pupil diameter. Mean time course of pupil diameter by subject.

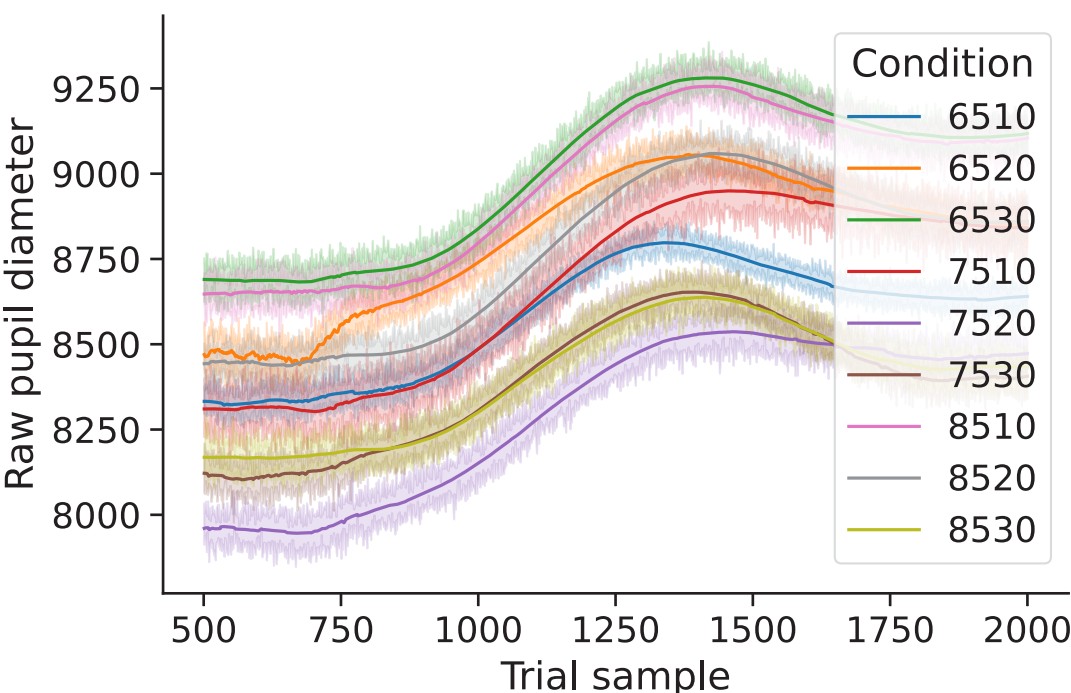

**Appendix 1—figure 10.** Evoked pupil diameter by condition. Mean time course of pupil diameter by condition.

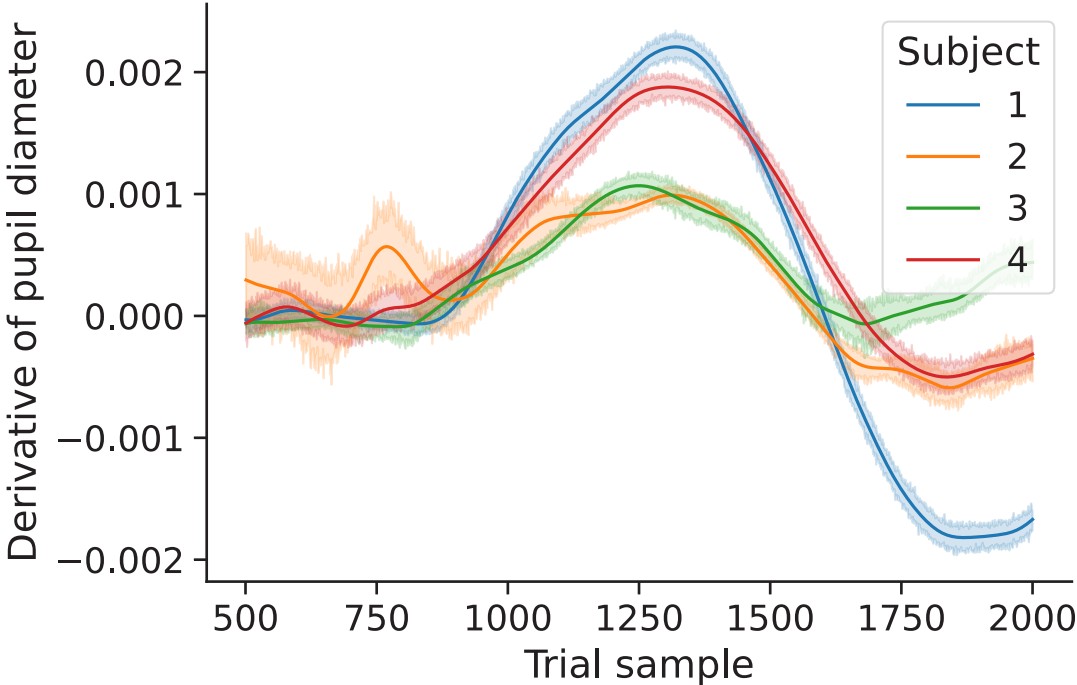

**Appendix 1—figure 11.** First temporal derivative of pupil diameter. Mean time course of derivative of pupil diameter by subject.

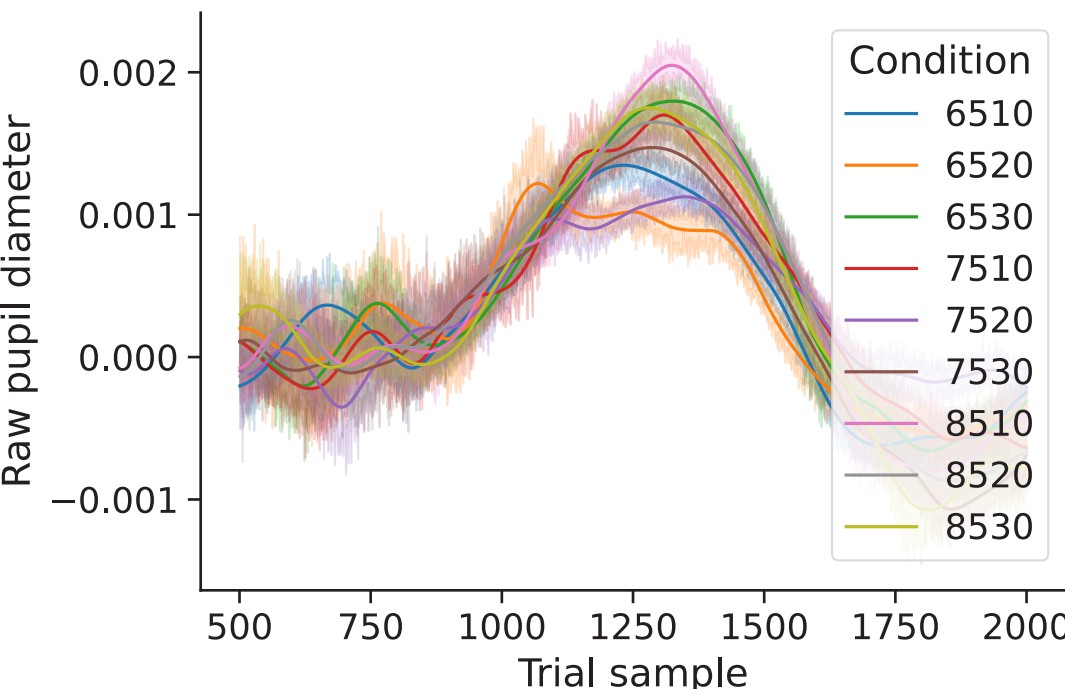

**Appendix 1—figure 12.** Derivative of evoked pupil diameter by condition. Mean time course of derivative of pupil diameter by condition.

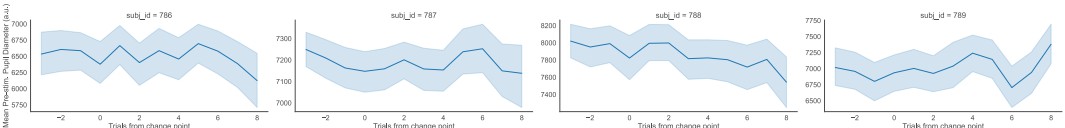

**Appendix 1—figure 13.** Prestimulus pupillary response. Mean prestimulus pupillary response by subject.

# Appendix 2

**Appendix 2—table 1.** Power analysis for Experiment 1.

The results of the model comparison analysis using simulated data. Roman numerals refer to a given model, as defined by the mapping between the ideal observer estimates and decision parameters in the first two columns. The column labeled DIC gives the raw DIC score, $\Delta\text{DIC}_{null}$ lists the change in model fit from an intercept-only model (the null-adjusted fit), and $\Delta\text{DIC}_{best}$ provides the change in null-adjusted model fit from the best-fitting model. The last row represents the null, intercept-only regression model. The best performing model is denoted by an asterisk.

| | $B$ | $\Omega$ | DIC | $\Delta\text{DIC}_{\text{null}}$ | $\Delta\text{DIC}_{\text{best}}$ |
|---|---|---|---|---|---|
| I* | $v$ | $a$ | −101886.4 | −15477.5 | 0.0 |
| II | $a$ | $v$ | −87486.7 | −1077.8 | 14399.7 |
| III | − | $v$ | −87373.9 | −965.0 | 14512.5 |
| IV | $v$ | − | −97634.70 | −11225.8 | 4251.70 |
| V | − | $a$ | −90577.3 | −4168.40 | 11309.00 |
| VI | $a$ | − | −86525.7 | −116.70 | 15360.7 |
| VII | − | − | −86408.9 | 0.0 | 15477.5 |

## Appendix 3

**Appendix 3—table 1.** Raw model selection results for Experiment 2.

The raw results of the model comparison analysis conducted depicted in *Table 1* for Experiment 2. Roman numerals refer to a given model, as defined by the mapping between the ideal observer estimates and decision parameters in the first two columns. The column labeled DIC gives the raw DIC score, $\Delta\text{DIC}_{null}$ lists the change in model fit from an intercept-only model (the null-adjusted fit), and $\Delta\text{DIC}_{best}$ provides the change in null-adjusted model fit from the best-fitting model. The last row for each subject represents the null, intercept-only regression model. Equivocal winning models marked with an asterisk and a tilde.

| Subject | $\Delta B$ | $\Omega$ | DIC | $\Delta DIC_{null}$ | $\Delta DIC_{best}$ |
|---|---|---|---|---|---|
| 1 | v | a | 5286.0 | −156.1 | 1.8 *sim |
| 1 | a | v | 5430.2 | −11.9 | 146.0 |
| 1 | - | v | 5431.3 | −10.8 | 147.1 |
| 1 | v | - | 5284.1 | −157.9 | 0.0 *sim |
| 1 | - | a | 5444.0 | 2.0 | 159.9 |
| 1 | a | - | 5441.1 | −0.9 | 157.0 |
| 1 | - | - | 5442.0 | 0.0 | 157.9 |
| 2 | v | a | 5162.9 | −144.1 | 0.0 *sim |
| 2 | a | v | 5283.0 | −24.1 | 120.0 |
| 2 | - | v | 5281.0 | −26.1 | 118.0 |
| 2 | v | - | 5165.1 | −142.0 | 2.1 *sim |
| 2 | - | a | 5303.7 | −3.4 | 140.8 |
| 2 | a | - | 5309.2 | 2.1 | 146.2 |
| 2 | - | - | 5307.1 | 0.0 | 144.1 |
| 3 | v | a | 3034.8 | −53.4 | 0.7 *sim |
| 3 | a | v | 3090.1 | 1.9 | 56.0 |
| 3 | - | v | 3089.5 | 1.2 | 55.4 |
| 3 | v | - | 3034.1 | −54.1 | 0.0 *sim |
| 3 | - | a | 3089.2 | 0.9 | 55.1 |
| 3 | a | - | 3088.8 | 0.6 | 54.7 |
| 3 | - | - | 3088.2 | 0.0 | 54.1 |
| 4 | v | a | 5438.9 | −7.7 | 1.4 *sim |
| 4 | a | v | 5450.5 | 3.8 | 13.0 |
| 4 | - | v | 5448.5 | 1.8 | 11.0 |
| 4 | v | - | 5437.5 | −9.1 | 0.0 *sim |
| 4 | - | a | 5448.2 | 1.6 | 10.7 |
| 4 | a | - | 5448.7 | 2.0 | 11.2 |
| 4 | - | - | 5446.7 | 0.0 | 9.1 |

