## [Editor Report]

The authors conducted an impressive study investigating dynamic adjustments in decision policies as a function of two types of uncertainty: decision conflict and volatility (change point probability). They combine learning model parameters with drift diffusion modeling to assess how the policy (as a combination of drift rate and threshold) varies with uncertainty and also test how these adjustments relate to the LC-NE system via pupil diameter. This work is impressive and will certainly be of interest to many.

---

## [Decision Letter]

**Decision letter after peer review:**

Thank you for submitting your article "Dynamic decision policy reconfiguration under outcome uncertainty" for consideration by *eLife*. Your article has been reviewed by 3 peer reviewers, one of whom is a member of our Board of Reviewing Editors, and the evaluation has been overseen by Floris de Lange as the Senior Editor. The following individuals involved in review of your submission have agreed to reveal their identity: Redmond O'Connell (Reviewer #1), Niels A Kloosterman (Reviewer #3).

Essential revisions:

1) The authors should rewrite sections of the manuscript to more clearly articulate for the reader the core theoretical questions that this study sets out to address. Beyond linking elements of choice uncertainty to parameters of an influential decision model, what were the study goals and why are they important/interesting? Similarly, the authors should clarify the broader implications of their findings for current theory and future research. In addition, the Results section is very long and at times it is hard to follow the rationale for each analysis step or how it relates to the study goals. The authors might consider moving certain details that are not essential to the Methods or Supplemental Materials.

2) The reviewers agreed that the DDM analyses seem overly restricted, being limited to only two parameters when there are other parameters of this model that can plausibly mediate the observed uncertainty effects. The authors make strong claims about the specific nature of the decision policy adjustments observed here but this interpretation would be greatly strengthened if the authors examined model variants that leave other parameters (e.g. non-decision time, drift bias). Please also provide some illustration of the fits to individual data.

3) Several concerns were raised regarding the pupillometry analyses.

a) First, there is a concern that any underlying relationships with choice strategy may have been obscured if the task stimuli evoked strong pupil light reflexes (constrictions) within the measurement window. This cannot be properly assessed without the authors providing plots of the average pupil diameter timecourses along with details regarding the stimuli (e.g. brightness).

b) Second, the authors appear to overlook several relevant metrics. This includes unbaselined prestimulus pupil diameter which has been linked to variations in choice performance in a number of investigations and also to exploration/exploitation switches in at least one report (Jepma and Nieuwenhuis 2011). Previous work has also shown that the LC-NE system is in fact better tracked by using the first time derivative of the pupil signal (Reimer et al., Nat Comm, 2016). The authors should consider looking at the pupil derivative to see if this reveals a link to their experimental manipulations. Importantly, using the derivative instead of the actual pupil time series attenuates the pupil light reflect since only the slope is taken.

c) Third, the authors should check whether they can replicate the relationships between uncertainty and pupil diameter that have been reported in the previous literature. This would go a long way toward confirming the validity both of their pupil data and the null result in the relationship with exploration/exploitation

The authors should also take note of the individual comments of the reviewers provided below as many additional suggestions are provided that should help further improve the manuscript.

*Reviewer #1:*

In this paper the authors seek to uncover the decision policy adjustments that underpin participants choices on a two-armed bandit task in which the relative reward and probability of reward associated with each alternative varied unpredictably over time. In an extensive modelling analysis the authors examined whether decision dynamics on this task could be understood in terms of adjustments to the bound and/or drift rate parameters of the drift diffusion model. The results indicate that when participants detect a change in which of two choice alternatives is most rewarding they switch from an exploitative to an exploratory decision strategy by lowering both the bound and drift rate of the decision process. My sense is that these findings are supported by extensive analyses reported in the paper.

1. I note that the authors allowed only one decision model parameter to map on to volatility and conflict. It would be interesting to know whether or not allowing either bound or drift rate to account for both volatility and conflict would improve the model fits.

2. The paper also reports a failure to detect any relationship between pupillary responses (here used as a proxy for noradrenergic arousal) during decision making and the aforementioned decision policy adjustments. Here the authors should cite the work of Joshi et al. (2016, Neuron) which, to my knowledge, was the first peer-reviewed paper to report a relationship between locus coeruleus activity and pupil diameter. This paper is also important to consider because they report a failure to observe the tonic/phasic firing modes originally reported by Aston-Jones and colleagues that form the basis for the present hypotheses.

3. The prior literature suggests that there are important functional distinctions between average absolute (i.e. unbaselined) pupil diameter measured in a given time window and the pupil dilation responses that are elicited during decision making. The authors do not consider the former which has been linked to exploration/exploitation strategies at least once in the prior literature (Jepma and Nieuwenhuis, 2011, J Cog Neuro) and which has been linked to variations in choice performance on several occasions (e.g. Murphy et al. 2011, Psychophysiology; Van Kempen et al. 2019, *eLife*). The authors then conducted a principal component analysis on 7 different metrics extracted from the stimulus-evoked pupil dilation response.

4. Aside from the desire to reduce dimensionality a clear rationale for this approach is not provided. This limits the ability to compare the present results to those in the related literature since most previous studies have investigated relationships between choice behaviour and metrics like pupil dilation amplitude, peak latency and onset latency individually.

5. The manuscript would benefit from more clearly articulating the theoretical advances/insights that it provides. It could be argued that the modelling work results in a situation where one set of cognitive constructs (change point detection and value estimation) are swapped for another set (bound and drift rate) but it is not clear what new understanding is gained from this.

*Reviewer #2:*

In the present study the authors investigated decision-policy adjustments as a function of two distinct forms of uncertainty, conflict and volatility. They extend previous studies on explore-exploit dynamics by investigating specifically how choice parameters in an evidence accumulation framework vary with uncertainty.

To that aim they experimentally manipulated conflict and volatility in a two armed bandit task and combined bayesian models of learning with evidence accumulation models of decision-making. Then they quantify the degree to which the relationship between estimated choice parameters – indexing the choice policy – varies as a function of the context in which the choice is made. They further recorded pupil diameters to index LC-NE activity and test whether policy adjustments are related to this activity

Choice conflict modulated the rate of evidence accumulation and change points – while also increasing conflict – decreased the boundary height, leading to fast exploratory choices. The authors show that choice dynamics are linked to uncertainty dynamics by driving systematic changes in the decision-policy characterized by the combination of drift rate and threshold. Fast increases in uncertainty following change points drove fast exploratory choices (low drift rate, low threshold) that gradually recovered to exploitatory choices as the new reward contingencies were learned (high drift rate, high threshold).

They further find no evidence that these changes relate to fluctuations in the LC-NE system as indexed with pupil diameter.

This work is methodologically very impressive and theoretically very interesting. It expands the space of decision-policies beyond the explore-exploit dichotomy and characterizes elegantly how decision-policies should dynamically vary along a two-dimensional policy space as the environment is found to change.

Overall, I really liked the study. I do however have some questions and suggestions for additional analyses and edits.

Questions:

1. Typically more similar options/conflict leads to longer RTs. In Exp 1 the authors find no effect and in Exp 2 the opposite. That's usually a pretty robust effect. Any idea why that's not the case here?

2. Also how does this square with the positive effect of ΔB on drift rate? I think that might be worth picking up and unpacking a bit, if only to show the superiority of model-based analyses to raw behavior given that the behavioral findings are super confusing and the parameter results make perfect sense.

3. Could that be some power issue (re number of subjects, not observations)? Is maybe one subject weird in exp. 2 and can't detect change points so well or track the values?

4. What is meant by the similar time course of belief updating for high and low volatility? (p 5 line 178) Shouldn't people update more under higher volatility? Is that what's captured in the change point probability parameter? Maybe that sentence could be clarified so it doesn't confuse readers familiar with work linking volatility and the α parameter in RL (as in more learning/updating under high volatility).

5. If I understand correctly, when change point probability goes up, ΔB always goes down. What's the correlation between the two and if there is a correlation, what does that mean for the impact of those learning parameters on decision parameters? Can you assess conflict effects independent of change point effects (are there period where values are stable and super similar)?

Suggestions:

6. Provide a clearer rationale for the model comparisons to test hypotheses about policy adjustments.

I honestly found it a bit difficult to follow the model comparison for theta. I also think that when the pupil is added, the rationale could be explained a bit more clearly to allow the reader to follow. Specifically I got confused about the intercept and time null models (is that just time or time relative to change point if the latter, why is that a null model?).

7. Replicate relationship between uncertainty parameters and pupil measures before linking it to policy adjustments.

As I understand, you find that the adjustments in the decision-policy are unrelated to pupil diameter. As a sanity check, have the authors looked at pupil diameter as a function of the uncertainty parameters? It would be good to show that earlier effects of uncertainty and their temporal dynamics are replicated (have a plot with the betas over time pre-choice and post feedback). I think the conclusion can be stronger if the authors show that pupil dilation tracks both forms of uncertainty and the anticipation and response dynamics associate with that, but that the subsequent adjustments are not mediated by this system.

Right now something could be wrong with the pupil data and the reader has no way to know. It would also be important just to see that these earlier findings replicate.

8. Reconsider causal language/interpretation of drift rate effects in the discussion.

You say in the discussion that people reduce the drift rate. Isn't the drift rate here determined by the consistency of the evidence? Sure, people can focus more (i.e. in cognitive control tasks, where the response rules are well known and errors are primarily driven by early incorrect activations of prepotent responses or attentional lapses), but I can focus all I want when there's no evidence (when I just don't know what the relative values are because I currently have zero [or little] valid experience to draw from after I detected a change point ) and my drift rate will still be low. No? Couldn't it be that participants have a sense that their drift rate is low (because they have no idea what's going on) and because taking time to sample would be useless (because uncertainty is not reducible other than through action), dropping the threshold is the right thing to do? In that sense the (expected) drift rate would dictate the optimal boundary height. I'm thinking of work by Tajima et al.

9. Reconsider reinterpretation of previous findings in the discussion – add nuance where nuance is due.

I have a bit of a problem with the authors’ assertion that previous findings relating boundary height and conflict could be a misattribution of volatility effects (Frank and colleagues). These previous studies did not have change points. So that is an unlikely explanation of that finding. What is more likely is that the choice dynamics were different because the choices were not temporally dependent, i.e. participants made choices between different options on each trial, meaning that the conflict and thus the optimal decision-strategy differed on every trial (in addition to any learning related uncertainty, but importantly, the true values associated with stimuli never changed). That is not the same as a change point/volatility. Further in the present study, conflict is anticipated, except in the case of change points. So that could equally be the difference between expected and unexpected uncertainty that leads to dissociable effects on decision strategies. In both cases, what drives the threshold adjustment is probably some form of surprise (unexpected conflict). As it stands, the statement in the discussion is inaccurate/misleading. That’s an easy fix though.

*Reviewer #3:*

Shifting between more explorative and more exploitative modes of decision making is crucial for adaptive human behavior. Therefore, the authors' attempt to investigate the internal processes that allow these modes is important to begin to understand this remarkable ability. In addition, investigating the proposed link to the LC-NE system is sensible and establishing its role in these processes would help the field forward. The authors present a thorough, modelling-heavy set of analysis on two interesting datasets aimed at revealing the underlying mechanisms.

1. Despite these strong points, the manuscript in its current version falls somewhat short of answering the questions that it poses. For one, the DDM analyses are restricted to only two parameters, which begs the question whether other established parameters might be better able to explain the results and thereby shed more light on the underlying mechanisms. Also, regarding the role of the pupil-linked LC-NE system, no strong conclusions can be drawn from the data, since the visual stimulus design likely resulted in strong pupil light reflexes, which might well have overshadowed subtler, more interesting modulations of the pupil. Despite the manuscripts innovative and clever use of Bayesian modelling and PCA, these two shortcomings might limit the impact of the manuscript in its current form on the field.

Shifting between more explorative and more exploitative modes of decision making is crucial for adaptive human behavior. Therefore, the authors' attempt to investigate the internal processes that allow these modes is important to begin to understand this remarkable faculty. In addition, investigating the proposed link to the LC-NE system is sensible and establishing its role in these processes would help the field forward. However, although in general the presented analyses seem thorough, I have two main concerns that in my opinion should be addressed before conclusions can be drawn from the data.

First, the DDM modelling is too restrictive, only focusing on the bound and drift parameters. Besides these two main parameters, another main parameter of the standard DDM is non-decision time, which to my surprise is not mentioned at all in the manuscript. Moreover, recent work has shown that two further parameters can capture internal processes possibly related to explore/exploit policies: starting point (z) and drift bias (called drift criterion or dc by Ratcliff and McKoon (2008)). Including these latter two parameters possibly can explain the RTs better than drift only and shed more light on the components underlying conflict and volatility. In addition, non-decision time might also be affected by the experimental manipulations, and should at least be reported in the manuscript (I assume that the authors did include it in their currently reported DDMs). In my mind, investigating all these further parameters is crucial before the conclusion that bound and drift rate best capture conflict and volatility is warranted.

My second point concerns the pupil analysis.

a) Although I could not find information about visual stimulus size and brightness in the methods, Figure 2AB suggests that there were strong visual transients at trial onset (black screen → stimulus), which presumably resulted in strong pupil constrictions due to the pupil light reflex (PLR).

b) I would have liked to see pupil time courses in this manuscript. The first components of the PCA, as employed by the authors (which in principle I think is a great idea) is likely to capture exactly these PLR dynamics given the large variance due to PLR.

c) Now, previous work has shown that the LC-NE system is in fact better tracked by using the first time derivative of the pupil signal (Reimer et al., Nat Comm, 2016). The authors should consider looking at the pupil derivative to see if this reveals a link to their experimental manipulations. Importantly, using the derivative instead of the actual pupil time series attenuates the PLR since only the slope is taken. Hence, when using the derivative, the PCA might pick up more interesting, cognitive drivers of pupil dynamics, since the PLR dynamics are suppressed. It would be interesting to see if this would reveal a link to the experimental manipulations.

d) Further, please note that the pupil likely not only is linked to the LC-NE system, but generally to catecholamines, which includes dopamine (Joshi et al. Neuron 2015). Therefore, I would recommend to not exclusively link pupil to LC-NE in the manuscript while interpreting the pupil results.

e) In any case, the author should show raw pupil as well as pupil derivative time courses for the different conditions to give insight in their data.

---

## [Author Response]

Essential revisions:1) The authors should rewrite sections of the manuscript to more clearly articulate for the reader the core theoretical questions that this study sets out to address. Beyond linking elements of choice uncertainty to parameters of an influential decision model, what were the study goals and why are they important/interesting? Similarly, the authors should clarify the broader implications of their findings for current theory and future research. In addition, the Results section is very long and at times it is hard to follow the rationale for each analysis step or how it relates to the study goals. The authors might consider moving certain details that are not essential to the Methods or Supplemental Materials.

We have made substantial changes to the frontend of manuscript (Abstract and Introduction) to more explicitly clarify the core theoretical questions that our study set out to address. We have also edited the Results and Discussion in targeted ways to better align the framing of our findings to the context of our theoretical questions. We hope that this better articulates the goals of our work.

2) The reviewers agreed that the DDM analyses seem overly restricted, being limited to only two parameters when there are other parameters of this model that can plausibly mediate the observed uncertainty effects. The authors make strong claims about the specific nature of the decision policy adjustments observed here but this interpretation would be greatly strengthened if the authors examined model variants that leave other parameters (e.g. non-decision time, drift bias). Please also provide some illustration of the fits to individual data.

The reviewers raise crucial points regarding the restricted set of model comparisons. Our original focus on these parameters was driven by prior work showing adaptation of the drift rate and boundary height terms under uncertainty, including findings linking cortico-basal ganglia dynamics with these drift-diffusion parameters under uncertain conditions (Dunovan et al. 2019; Dunovan and Verstynen 2019; Rubin et al. 2021). As we detail below, we have now expanded the set of models considered to include other plausible decision parameters (non-decision time (tr), drift criterion / drift bias (dc), and the starting point (z), in addition to the drift rate (v) and boundary height (a)).

These more rigorous analyses have resulted in an update to the change-point-evoked effect on boundary height (*a)*. Instead of *a decreasing* in response to a suspected change in reward contingencies, *a increases* in response to a suspected change. Ironically, this is consistent with our preregistered
hypothesis as well as with prior experimental results in our lab (and others). We have updated both our key figures and the text of the Results and Discussion to reflect this change. (See response below).

3) Several concerns were raised regarding the pupillometry analyses.a) First, there is a concern that any underlying relationships with choice strategy may have been obscured if the task stimuli evoked strong pupil light reflexes (constrictions) within the measurement window. This cannot be properly assessed without the authors providing plots of the average pupil diameter timecourses along with details regarding the stimuli (e.g. brightness).

We were careful to control the effect of light on the pupillary response during data collection. This included the construction of a specific rig around the testing computer so as to reduce ambient reflection of light from walls and ceiling.

To control luminance we also used a Derrington-Krauskopf-Lennie (DKL) color space that allows for direct luminance control. As a result, the stimulus presentation display was rendered isoluminant throughout the task. In addition, the lead author built a booth to isolate the participant from ambient sources of light during data collection.

We now mention these details in the Methods:

“Throughout the task, the head-stabilized diameter and gaze position of the left pupil were measured with an Eyelink 1000 desktop mount at 1000 Hz. […] During the reward-learning task, we used this method to isolate the task-evoked pupillary response.”

Additionally, we include a reminder as part of the caption for Figure 2B to aid the reader in interpreting the results as they progress through the paper:

“(B) In Experiment 2, participants were asked to choose between one of two Greebles (one male, one female). The total number of points earned was displayed at the center of the screen. The stimulus display was rendered isoluminant throughout the task.”

(b) Second, the authors appear to overlook several relevant metrics. This includes unbaselined prestimulus pupil diameter which has been linked to variations in choice performance in a number of investigations and also to exploration/exploitation switches in at least one report (Jepma and Nieuwenhuis 2011). Previous work has also shown that the LC-NE system is in fact better tracked by using the first time derivative of the pupil signal (Reimer et al., Nat Comm, 2016). The authors should consider looking at the pupil derivative to see if this reveals a link to their experimental manipulations. Importantly, using the derivative instead of the actual pupil time series attenuates the pupil light reflect since only the slope is taken.

We have reanalyzed the data using un-baselined prestimulus pupil diameter and the first time derivative of the pupillary response. We again observe a null result. These analyses are detailed as part of our point-by-point reply to reviewers below. Figures visualizing these time courses have been added to the supplementary section of the manuscript (Supplementary Figures 10 and 11).

c) Third, the authors should check whether they can replicate the relationships between uncertainty and pupil diameter that have been reported in the previous literature. This would go a long way toward confirming the validity both of their pupil data and the null result in the relationship with exploration/exploitation

We have now conducted analyses to check for the previously reported links between exploratory choice behavior and the pupillary response. However, we observe no clear evidence for these pre-established links. Therefore, we qualify the use of the pupillary data. The Results section now states this lack of replication to titrate the reader’s confidence in the pupillary results and the corresponding inferences relating to LC-NE system influence on the progression through the decision manifold:

“Specifically, if the LC-NE system were sensitive to a change in the optimal choice, then we should observe a moderate spike in phasic activity following a change in action-outcome contingencies. […] We ask the reader to titrate their interpretation of these pupillary data accordingly.”

We include a similar addition to the pupillometry subsection of the Methods:

“Note that we also conducted a similar analysis using more conventional methods to assess the task-evoked pupillary response and observed another null effect. […] As such, we caution the reader to view our pupillary results in light of this lack of replication of pre-established exploration-driven pupillary responses.”

Reviewer #1:In this paper the authors seek to uncover the decision policy adjustments that underpin participants choices on a two-armed bandit task in which the relative reward and probability of reward associated with each alternative varied unpredictably over time. In an extensive modelling analysis the authors examined whether decision dynamics on this task could be understood in terms of adjustments to the bound and/or drift rate parameters of the drift diffusion model. The results indicate that when participants detect a change in which of two choice alternatives is most rewarding they switch from an exploitative to an exploratory decision strategy by lowering both the bound and drift rate of the decision process. My sense is that these findings are supported by extensive analyses reported in the paper.1. I note that the authors allowed only one decision model parameter to map on to volatility and conflict. It would be interesting to know whether or not allowing either bound or drift rate to account for both volatility and conflict would improve the model fits.

The choice to map distinct ideal observer estimates to distinct decision parameters reflects the theoretical motivation underlying the development of our hypotheses tested here. We now expand on this in the introduction to make clear the reasoning for our narrow focus, as below:

“Are the parameters that govern accumulation of evidence for decision making modifiable? […] These policies, in turn, adaptively reconfigure based on current environmental feedback signals by modulating value estimation and the rate of selection errors (Figure 1E).”

A complete, exhaustive sweep of decision parameters as proposed would be computationally inefficient. This would require evaluation of all possible single parameter, dual-parameter (*n*-parameter) pairings, with both many-to-one ideal observer to DDM mappings and many-to-one DDM parameter mappings to ideal observer mappings. As model complexity increases in these hierarchical DDM fits, the convergence or stability of the fits is more difficult to achieve. So we opted for parsimony in the set of model fits to avoid a data mining expedition that would very likely lead to a set of inconclusive model fits on overly complex models. This sort of restricted set test is quite common when using hierarchical DDM (and, indeed, many hierarchical models in general). Further, estimating pairwise ideal observer to DDM parameter mappings alone keeps model complexity constant, allowing us to make clear comparisons in information loss scores between candidate models.

2. The paper also reports a failure to detect any relationship between pupillary responses (here used as a proxy for noradrenergic arousal) during decision making and the aforementioned decision policy adjustments. Here the authors should cite the work of Joshi et al. (2016, Neuron) which, to my knowledge, was the first peer-reviewed paper to report a relationship between locus coeruleus activity and pupil diameter. This paper is also important to consider because they report a failure to observe the tonic/phasic firing modes originally reported by Aston-Jones and colleagues that form the basis for the present hypotheses.

We thank the reviewer for this omitted reference. Indeed, Joshi et al. 2016 provides clear evidence of a link between locus coeruleus activity and pupil diameter. To our knowledge, this observation extends back to the work of Rajkowski, Kubiak, and Aston-Jones 1994, showing the phasic and tonic modes of the locus-coeruleus system in relation to exploratory behavior. All of this prior work is clearly important to consider, and we thank the reviewer for bringing this to our attention.

We now cite both studies in our revised manuscript:

“The LC-NE system is known to modulate exploration states under uncertainty and pupil diam- eter shows a tight correspondence with LC neuron firing rate (Aston-Jones and Cohen, 2005; Rajkowski et al., 1994), with changes in pupil diameter indexing the explore-exploit decision state (Jepma and Nieuwenhuis, 2011). Similar to the classic Yerkes-Dodson curve relating arousal to performance (Yerkes et al., 1908), performance is optimal when tonic LC activity is moderate and phasic LC activity increases following a goal-related stimulus (Aston-Jones et al. (1999), but see Joshi et al. (2016) for an exception).”

3. The prior literature suggests that there are important functional distinctions between average absolute (i.e. unbaselined) pupil diameter measured in a given time window and the pupil dilation responses that are elicited during decision making. The authors do not consider the former which has been linked to exploration/exploitation strategies at least once in the prior literature (Jepma and Nieuwenhuis, 2011, J Cog Neuro) and which has been linked to variations in choice performance on several occasions (e.g. Murphy et al. 2011, Psychophysiology; Van Kempen et al. 2019, eLife).

Thank you for bringing this important functional distinction between un-baselined pupil diameter and the dilation response to our attention. In our data, if baseline pupil diameter were sensitive to shifts from exploitation to exploration, then we should observe a change in baseline pupil diameter proximal to a change point. However, we do not observe a change-point-evoked shift in baseline pupil diameter in our data, as we might expect given the previous links to exploratory behavior. We now mention our lack of support for these validation analyses in the Results section and visualize the pupillary time courses and results in the Supplementary section (Supp. Figures 10-13).

“Specifically, if the LC-NE system were sensitive to a change in the optimal choice, then we should observe a moderate spike in phasic activity following a change in action-outcome contingencies. Note that we do not observe previously established links between exploratory choice behavior and the pupillary response (Jepma and Nieuwenhuis, 2011; Murphy et al., 2011; van Kempen et al., 2019). We ask the reader to titrate their interpretation of these pupillary data accordingly.”

4. The authors then conducted a principal component analysis on 7 different metrics extracted from the stimulus-evoked pupil dilation response. Aside from the desire to reduce dimensionality a clear rationale for this approach is not provided. This limits the ability to compare the present results to those in the related literature since most previous studies have investigated relationships between choice behaviour and metrics like pupil dilation amplitude, peak latency and onset latency individually.

From a computational perspective, reducing the dimensionality of this set of pupillary response metrics expands the set of models we can consider without taxing computational resources in a reasonable amount of time.

Further, our original PCA method was intended to maximize the variability of the pupillary response linked to the decision manifold. This allowed us to capture separable sources of variance relating to timing and amplitude effects without restricting the data to a smaller set of metrics and possibly discarding information (e.g. timing effects may not be constrained to peak latency or onset latency; amplitude effects may not be constrained to peak dilation amplitude).

We have edited the Results section with the motivation for our PCA approach:

“We characterized the evoked pupillary response on each trial using seven metrics: the mean of the pupil data over each trial interval, the latency to the peak onset and offset, the latency to peak amplitude, the peak amplitude, and the area under the curve of the pupillary response. […] Therefore, we submitted these metrics to principal component analysis to reduce their dimensionality while capturing maximum variance.”

We have also reanalyzed these pupillary data using conventional analysis methods and continue to observe a null effect (see points 3C and 7 for Reviewer 3). We have edited the Results section to reflect this:

“Thus, for interpretability, we refer to the first and second principal components as timing and magnitude components, respectively (Figure 9B). Note that we also conduct this analysis using more conventional methods of pupillary analysis and continue to observe a null effect (see the Pupil data preprocessing for details).”

5. The manuscript would benefit from more clearly articulating the theoretical advances/insights that it provides. It could be argued that the modelling work results in a situation where one set of cognitive constructs (change point detection and value estimation) are swapped for another set (bound and drift rate) but it is not clear what new understanding is gained from this.

This is an excellent point and it reflects our somewhat opaque framing in the

Introduction, which we have now fixed (see our response to the first point raised by the Review Editor). Our primary goal was to understand how the evidence accumulation dynamics changed when the environment requires reassessing learned state action values. Rather than think of these accumulation dynamics, driven by drift rate, boundary height, etc., as a static process, we make the case of thinking of these as points on a continuum of possible states (see manifold in Figure 1e). So our primary focus is on the algorithms of information processing. However, if these are dynamic processes, e.g., drift rate fluctuates over time, then there has to be some learning signal that drives these fluctuations. We chose the ideal observer parameters as likely learning signals that drive plasticity in the decision policy state. We acknowledge that these may not be the *only* signals that drive adjustments in decision policy dynamics, but they are ideal in that they reflect two correlated, but separate estimates of environmental state.

In line with our response to point 1 from Reviewer 1, we now make this clearer in the Introduction:

“Knowing how decision policies shift in the face of dynamic environments requires looking at the algorithmic properties of the policy itself. […] We predicted that, in response to suspected changes in action-outcome contingencies, humans would exhibit a stereotyped adjustment in the drift rate and boundary height that pushes decisions from certain, exploitative states to uncertain, exploratory states and back again (Figure 1E).”

Reviewer #2:[…] Overall, I really liked the study. I do however have some questions and suggestions for additional analyses and edits.Questions:1. Typically more similar options/conflict leads to longer RTs. In Exp 1 the authors find no effect and in Exp 2 the opposite. That's usually a pretty robust effect. Any idea why that's not the case here?

The reviewer is correct, we do observe different effects of conflict on reaction time in Experiments 1 and 2. The effect is absent in Experiment 1 and small enough in Experiment 2 that it can effectively be considered a null finding. One possible reason for attenuated effects of conflict on reaction time is that participants were overtrained

(Experiment 1 was 2hrs per participant, Experiment 2 was nine hours per participant). This may result in participants having developed an expectation of change point frequency and/or conflict manipulation. If this were the case, then we might expect to see an effect on reaction times in Experiment 1, where participants undergo four sessions of training, but not in Experiment 2, where participants undergo nine sessions of training each. However, we instead see negligible effects on reaction times in Experiment 2 and no effect of conflict on reaction times in Experiment 1.

A second possibility relates to the complexity of our manipulations. Here we impose a range of conflict levels that also vary with degrees of volatility, meaning that our observed reaction time effects are a mix of responses to different extents of conflict and volatility together. This contrasts with previous reports measuring a more restricted range of conflict and without the influence of volatility. Thus, participants are tracking two sources of uncertainty, which may attenuate the overall observed RT response to conflict alone.

We now acknowledge this discrepancy in the Discussion.

“Previous literature has shown a conflict-induced spike in reaction time (e.g. Jahfari et al. 2019). […] Future research should explore the interaction of change point and conflict estimation on the speed-accuracy tradeoff.”

Either way, this discrepancy between our results and other studies, as well as the lack of internal replication of the change point probability and boundary height association in Experiment 2 of our results, is an interesting avenue of exploratory research that we are currently following up on.

2. Also how does this square with the positive effect of ΔB on drift rate? I think that might be worth picking up and unpacking a bit, if only to show the superiority of model-based analyses to raw behavior given that the behavioral findings are super confusing and the parameter results make perfect sense.

The reviewer is absolutely correct. The lack of simple main effects on RT across experiments obscures meaningful behavioral patterns that can be detected with a model-based approach. We now reference the value of a model-based analysis after reviewing the ambiguous behavioral results:

“At the gross level, across all trials within an experimental condition, increasing the ambiguity of the optimal choice (conflict) and increasing the instability of action outcomes (volatility) decreases the probability of selecting the optimal choice. […] We adopt a more focal, model-based analysis in the next section to clarify these peri-change point dynamics.”

3. Could that be some power issue (re number of subjects, not observations)? Is maybe one subject weird in exp. 2 and can't detect change points so well or track the values?

We see fairly consistent sensitivity to change points across subjects in Experiment 2, with accuracy plummeting and recovering with similar time courses, suggesting that all participants track the value of the optimal choice in a consistent manner (see Supplementary Figures 8 and 9 for evoked response profiles by subject). In addition, we conducted a power analysis prior to data collection (preregistration) and the within-session, within-subject power for Experiment 2 is still high enough to detect our hypothesized effects.

As in our response to the previous comment, we think that the RT effects are masking compensatory changes in two different parameters in the accumulation process, rather than outlier participants or sessions.

That said, it may be the case that the reaction time effects simply require more power to detect than accuracy effects, and both of our experiments fail to detect them for that reason. We plan to conduct a replication experiment to recover the effects we observed using a high-powered experimental design both within and across subjects. We hope that the results of this replication address this question. However, given that this is a tangential focus to our original research question, it is more suitable to address this as a follow up paper.

4. What is meant by the similar time course of belief updating for high and low volatility? (p 5 line 178) Shouldn't people update more under higher volatility? Is that what's captured in the change point probability parameter? Maybe that sentence could be clarified so it doesn't confuse readers familiar with work linking volatility and the α parameter in RL (as in more learning/updating under high volatility).

This was an awkwardly worded sentence. We apologize. What we meant here is that the rate of change in relative reward value (ΔB) after a change point is qualitatively similar under both low and high volatility conditions. In other words, the slope of the lines for the two volatility manipulations in Figure 4A is approximately the same. However, the change-point-evoked response belies the main effect of volatility on overall estimates of relative reward value, as shown in Figure 4B. Therefore, we removed this sentence to maintain clarity.

5. If I understand correctly, when change point probability goes up, ΔB always goes down. What's the correlation between the two and if there is a correlation, what does that mean for the impact of those learning parameters on decision parameters? Can you assess conflict effects independent of change point effects (are there period where values are stable and super similar)?

These two parameters are indeed correlated. In fact, ΔB is included in the calculation of change point probability and vice versa. But this interdependence is expected: your certainty in value is going to decrease if you live in a chaotic and changing world. However, the real question is whether they are too correlated to impact our model interpretability. The correlation between change point probability and ΔB is small but reliable, with an increase in belief as change point probability decreases (Spearman’s rho = -0.234 +/- 0.029). However, the Variance Inflation Factor, a collinearity metric, is within an acceptable range for both experiments (Experiment 1: 1.100 +/- 0.013; Experiment 2: 1.058 +/- 0.017). This is generally considered to be an acceptable degree of collinearity (values greater than 10 cause concern; Chatterjee and Simonoff 2013, p. 28-29, notebook showing these results). Thus, the degree of correlation between change point probability and ΔB should have a minimal effect on the estimation of the decision parameters. This suggests that we can safely estimate independent effects of volatility and conflict using change point probability and belief.

Suggestions:6. Provide a clearer rationale for the model comparisons to test hypotheses about policy adjustments.I honestly found it a bit difficult to follow the model comparison for theta. I also think that when the pupil is added, the rationale could be explained a bit more clearly to allow the reader to follow. Specifically I got confused about the intercept and time null models (is that just time or time relative to change point if the latter, why is that a null model?).

We thank the reviewer for bringing this lack of clarity to our attention. The time-null model tested for an impact of time relative to a change point, separate from conditional influences of volatility and conflict and the influence of the pupillary response.

We have renamed these models for clarity and expanded on our selection logic for models specifying an impact on decision policy adjustment and updated our naming convention in the Methods section and in the Results section:

“First, we tested the null hypothesis that the decision dynamics was solely a function of the intercept, or the average of the decision dynamics. […] We call this the evoked response model.”

7. Replicate relationship between uncertainty parameters and pupil measures before linking it to policy adjustments.As I understand, you find that the adjustments in the decision-policy are unrelated to pupil diameter. As a sanity check, have the authors looked at pupil diameter as a function of the uncertainty parameters? It would be good to show that earlier effects of uncertainty and their temporal dynamics are replicated (have a plot with the betas over time pre-choice and post feedback). I think the conclusion can be stronger if the authors show that pupil dilation tracks both forms of uncertainty and the anticipation and response dynamics associate with that, but that the subsequent adjustments are not mediated by this system.Right now something could be wrong with the pupil data and the reader has no way to know. It would also be important just to see that these earlier findings replicate.

The reviewer raises an excellent point. We did look into this relationship and yet failed to observe evidence for a relationship between our uncertainty parameters – belief and change point probability – and the pupillary response, as measured by both the metrics we calculated and the principal components derived from those metrics.

We have now stated this lack of replication in the Results section to caution the reader:

“Specifically, if the LC-NE system were sensitive to a change in the optimal choice, then we should observe a moderate spike in phasic activity following a change in action-outcome contingencies. […] We ask the reader to titrate their interpretation of these pupillary data accordingly and to view the corresponding inferences relating noradrenergic and catecholaminergic systems to decision policy adjustment in this light.”

8. Reconsider causal language/interpretation of drift rate effects in the discussion.You say in the discussion that people reduce the drift rate. Isn't the drift rate here determined by the consistency of the evidence? Sure, people can focus more (i.e. in cognitive control tasks, where the response rules are well known and errors are primarily driven by early incorrect activations of prepotent responses or attentional lapses), but I can focus all I want when there's no evidence (when I just don't know what the relative values are because I currently have zero [or little] valid experience to draw from after I detected a change point ) and my drift rate will still be low. No? Couldn't it be that participants have a sense that their drift rate is low (because they have no idea what's going on) and because taking time to sample would be useless (because uncertainty is not reducible other than through action), dropping the threshold is the right thing to do? In that sense the (expected) drift rate would dictate the optimal boundary height. I'm thinking of work by Tajima et al.

The reviewer brings up two points in this comment. We will address each separately.

First there appears to be a conflation of intention with causation. While we fully agree that we can reduce the causal certainty of the language used in the Discussion (something we now do in the revised text), the fact remains that in our data, drift rate reliably changes in response to a changepoint in a stereotypic fashion. Given the nature of the experimental design, we are careful to not make any assumptions as to whether this change is driven by explicit or intentional mechanisms (e.g., increased focus) versus implicit or automatic mechanisms.

The second point raised regards alignment with the work by Tajima and colleagues. It is very likely that the drift rate and boundary height are changing in a cooperative, adaptive manner, at least insofar as the trials immediately surrounding a change point are concerned. The temporal profile of the two parameter changes (at least in Exp. 1 given our new analysis) is quite different, with boundary height changes being brief and drift rate adaptation requiring more time. So, if a change in the boundary height is dictating drift rate changes it is only happening briefly. Therefore we think that a majority of the changes seen in response to a change point are occurring through independent means (consistent with our prior computational models of these pathways (Dunovan et al. 2019; Dunovan and Verstynen 2019; Rubin et al. 2021)).

9. Reconsider reinterpretation of previous findings in the discussion – add nuance where nuance is due.I have a bit of a problem with the authors' assertion that previous findings relating boundary height and conflict could be a misattribution of volatility effects (Frank and colleagues). These previous studies did not have change points. So that is an unlikely explanation of that finding. What is more likely is that the choice dynamics were different because the choices were not temporally dependent, i.e. participants made choices between different options on each trial, meaning that the conflict and thus the optimal decision-strategy differed on every trial (in addition to any learning related uncertainty, but importantly, the true values associated with stimuli never changed). That is not the same as a change point/volatility. Further in the present study, conflict is anticipated, except in the case of change points. So that could equally be the difference between expected and unexpected uncertainty that leads to dissociable effects on decision strategies. In both cases, what drives the threshold adjustment is probably some form of surprise (unexpected conflict). As it stands, the statement in the discussion is inaccurate/misleading. That's an easy fix though.

Thank you for this careful reading of our critique. Given the update to our results after the more thorough set of model comparisons requested, we no longer include this point in the Discussion. Further, we have made sure to qualify our interpretation of how our findings integrate with the broader literature where necessary. We hope this reflects a more nuanced view of the prior literature.

Reviewer #3:Shifting between more explorative and more exploitative modes of decision making is crucial for adaptive human behavior. Therefore, the authors' attempt to investigate the internal processes that allow these modes is important to begin to understand this remarkable ability. In addition, investigating the proposed link to the LC-NE system is sensible and establishing its role in these processes would help the field forward. The authors present a thorough, modelling-heavy set of analysis on two interesting datasets aimed at revealing the underlying mechanisms.1. Despite these strong points, the manuscript in its current version falls somewhat short of answering the questions that it poses. For one, the DDM analyses are restricted to only two parameters, which begs the question whether other established parameters might be better able to explain the results and thereby shed more light on the underlying mechanisms. Also, regarding the role of the pupil-linked LC-NE system, no strong conclusions can be drawn from the data, since the visual stimulus design likely resulted in strong pupil light reflexes, which might well have overshadowed subtler, more interesting modulations of the pupil. Despite the manuscripts innovative and clever use of Bayesian modelling and PCA, these two shortcomings might limit the impact of the manuscript in its current form on the field.Shifting between more explorative and more exploitative modes of decision making is crucial for adaptive human behavior. Therefore, the authors' attempt to investigate the internal processes that allow these modes is important to begin to understand this remarkable faculty. In addition, investigating the proposed link to the LC-NE system is sensible and establishing its role in these processes would help the field forward. However, although in general the presented analyses seem thorough, I have two main concerns that in my opinion should be addressed before conclusions can be drawn from the data.First, the DDM modelling is too restrictive, only focusing on the bound and drift parameters. Besides these two main parameters, another main parameter of the standard DDM is non-decision time, which to my surprise is not mentioned at all in the manuscript. Moreover, recent work has shown that two further parameters can capture internal processes possibly related to explore/exploit policies: starting point (z) and drift bias (called drift criterion or dc by Ratcliff and McKoon (2008)). Including these latter two parameters possibly can explain the RTs better than drift only and shed more light on the components underlying conflict and volatility. In addition, non-decision time might also be affected by the experimental manipulations, and should at least be reported in the manuscript (I assume that the authors did include it in their currently reported DDMs). In my mind, investigating all these further parameters is crucial before the conclusion that bound and drift rate best capture conflict and volatility is warranted.

The reviewer is correct. Investigating the remaining DDM parameters is crucial to substantiate our claim that the drift rate and boundary height respond in a coordinated fashion to promote exploration in response to a suspected change. This was something that we did in our initial model evaluations but was left out for the sake of concision. We now include a more thorough test of the set of DDM parameters (*a,v,t,z,dc*) that could respond to a change point. This broader first-level test confirms our initial results showing that the *t*, *z*, and *dc* parameters do not reliably change in response to a change point (Figure 5).

My second point concerns the pupil analysis. Regarding the role of the pupil-linked LC-NE system, no strong conclusions can be drawn from the data, since the visual stimulus design likely resulted in strong pupil light reflexes, which might well have overshadowed subtler, more interesting modulations of the pupil.a) Although I could not find information about visual stimulus size and brightness in the methods, Figure 2AB suggests that there were strong visual transients at trial onset (black screen → stimulus), which presumably resulted in strong pupil constrictions due to the pupil light reflex (PLR).

The representation of the display depicted in Figures 2A and B does not show the actual luminance of the stimulus display for Experiment 2. As we state in our general response to the Editor above and in point B of this response, we carefully controlled task-related luminance and ambient sources of light in order to maximize our capacity to detect subtle pupillary effects. See point 3A in our response to the Editor and the revised language included in that response.

b) I would have liked to see pupil time courses in this manuscript. The first components of the PCA, as employed by the authors (which in principle I think is a great idea) is likely to capture exactly these PLR dynamics given the large variance due to PLR.

It is unlikely that we are capturing pupillary light reflexes given our control of luminance in the experimental testing rig. However, we have now included average time courses of the pupillary response to the Supplementary section (Supp. Figure 9-13). We hope these are useful for readers who share this concern. See point 3C of our response to the editor for cautionary language added to the Results and Methods section.

c) Now, previous work has shown that the LC-NE system is in fact better tracked by using the first time derivative of the pupil signal (Reimer et al., Nat Comm, 2016). The authors should consider looking at the pupil derivative to see if this reveals a link to their experimental manipulations. Importantly, using the derivative instead of the actual pupil time series attenuates the PLR since only the slope is taken. Hence, when using the derivative, the PCA might pick up more interesting, cognitive drivers of pupil dynamics, since the PLR dynamics are suppressed. It would be interesting to see if this would reveal a link to the experimental manipulations.

We have now calculated the first time derivative of the pupil signal and reanalyzed our data on this measure. Using the first time derivative of the pupil signal, we recalculated the principal components of the pupillary response as with the first order signal. Using these recalculated principal components, we reassessed the relationship between the pupillary data and our conditional manipulations and retested the link between these principal components and theta, the relationship between *a* and *v*. We continue to observe null effects. See point 4 of our response to Reviewer 1.

d) Further, please note that the pupil likely not only is linked to the LC-NE system, but generally to catecholamines, which includes dopamine (Joshi et al. Neuron 2015). Therefore, I would recommend to not exclusively link pupil to LC-NE in the manuscript while interpreting the pupil results.

We appreciate the point that other catecholamines, such as dopamine, also contribute to the task-evoked pupillary response. We now acknowledge this lack of specificity in the Discussion:

“We hypothesized that these shifts in decision policies would be linked to changes in phasic responses of the LC-NE pathways, although we should note our experimental design does not distinguish between pupillary dynamics driven by other catecholamines, such as dopamine, and those dynamics driven by the LC-NE system.”

e) In any case, the author should show raw pupil as well as pupil derivative time courses for the different conditions to give insight in their data.

We now include subject-wise visualizations of the evoked pupillary response and the time derivative of that response for all combinations of conflict and volatility in the Supplementary section (Supp. Figures 11 and 13).